# Diversity and evolution of cerebellar folding in mammals

Katja Heuer[1]*, Nicolas Traut[1], Alexandra Allison de Sousa[2], Sofie Louise Valk[3,4,5], Julien Clavel[6], Roberto Toro[1]*

[1]Institut Pasteur, Université Paris Cité, Unité de Neuroanatomie Appliquée et Théorique, Paris, France; [2]Centre for Health and Cognition, Bath Spa University, Bath, United Kingdom; [3]Otto Hahn Group Cognitive Neurogenetics, Max Planck Institute for Human Cognitive and Brain Sciences, Leipzig, Germany; [4]Institute of Neuroscience and Medicine, Brain & Behaviour (INM-7), Research Centre Jülich, FZ Jülich, Jülich, Germany; [5]Institute of Systems Neuroscience, Medical Faculty, Heinrich Heine University Düsseldorf, Düsseldorf, Germany; [6]Université Lyon, Université Claude Bernard Lyon 1, CNRS, ENTPE, UMR 5023 LEHNA, Villeurbanne, France

**Abstract** The process of brain folding is thought to play an important role in the development and organisation of the cerebrum and the cerebellum. The study of cerebellar folding is challenging due to the small size and abundance of its folia. In consequence, little is known about its anatomical diversity and evolution. We constituted an open collection of histological data from 56 mammalian species and manually segmented the cerebrum and the cerebellum. We developed methods to measure the geometry of cerebellar folia and to estimate the thickness of the molecular layer. We used phylogenetic comparative methods to study the diversity and evolution of cerebellar folding and its relationship with the anatomy of the cerebrum. Our results show that the evolution of cerebellar and cerebral anatomy follows a stabilising selection process. We observed two groups of phenotypes changing concertedly through evolution: a group of 'diverse' phenotypes – varying over several orders of magnitude together with body size, and a group of 'stable' phenotypes varying over less than 1 order of magnitude across species. Our analyses confirmed the strong correlation between cerebral and cerebellar volumes across species, and showed in addition that large cerebella are disproportionately more folded than smaller ones. Compared with the extreme variations in cerebellar surface area, folial anatomy and molecular layer thickness varied only slightly, showing a much smaller increase in the larger cerebella. We discuss how these findings could provide new insights into the diversity and evolution of cerebellar folding, the mechanisms of cerebellar and cerebral folding, and their potential influence on the organisation of the brain across species.

*For correspondence:
katjaqheuer@gmail.com (KH);
rto@pasteur.fr (RT)

**Competing interest:** The authors declare that no competing interests exist.

## Editor's evaluation

This fundamental study gives novel insight into the folding diversity of the cerebellum compared to the cerebrum among 56 mammalian species. Based on impressive data, a variety of convincing analyses are performed, in particular for anatomical descriptions, phylogenetic comparisons and allometry investigations. This study will be of great interest to biologists, especially evolutionary and comparative neuroscientists, and physicists interested in biomechanics, as these observations provide a basis for models of brain folding mechanisms.

## Introduction

Brain folding may play an important role in facilitating and inducing a variety of anatomical and functional organisation patterns (*Welker, 1990*; *Toro, 2012*). Among the many folded structures of the mammalian brain, the cortices of the cerebrum and the cerebellum are the two largest ones. In humans, the cerebellum has ~80% of the surface area of the cerebral cortex (*Sereno et al., 2020*), and contains ~80% of all brain neurons, although it represents only ~10% of the brain mass (*Azevedo et al., 2009*). Both cortices are organised in layers, with neuronal cell bodies occupying a superficial position and principal neurons – pyramidal neurons in the cerebral cortex, Purkinje cells in the cerebellum – sending axons towards an internal white matter. Many aspects of the organisation, development, and evolution of the cerebral and cerebellar cortex are, however, very different. While the cerebral cortex is unique to mammals, the cerebellum is present in all vertebrates, with cerebellum-like structures which can be even identified in invertebrates such as octopuses (*Nieuwenhuys et al., 1998*; *Shigeno et al., 2018*). While the main target of pyramidal cells is the cerebral cortex itself (through profuse cortico-cortical connections), Purkinje cell afferents are mostly feedforward (*Ramnani, 2006*; *Nieuwenhuys et al., 2008*; *Yamamoto et al., 2012*). While dendritic trees of pyramidal neurons can be found in all cortical layers, those of Purkinje cells are restricted to the molecular layer – their nuclei forming a boundary between the molecular and granular layers. While the cortex is only folded in relatively large mammals, the cerebellum is folded in all of them, and is also folded in birds (*Cunha et al., 2020*; *Cunha et al., 2021*), some fish (*Yopak et al., 2007*) but not in reptiles (*Larsell, 1926*; *Nieuwenhuys et al., 1998*). It is interesting then to compare folding in both structures as a first step towards understanding how the mechanics of folding could influence the organisation in these two different types of tissue (*Franze, 2013*; *Kroenke and Bayly, 2018*; *Foubet et al., 2019*; *Heuer and Toro, 2019*).

Here, we aimed at characterising the folding of the cerebellar cortex using histological data from a sample of 56 mammalian species. The analysis of cerebellar folding is made difficult by the small size of the folia. Magnetic resonance imaging, the main tool for studying the folding of the cerebral cortex, does not provide the resolution required to distinguish and reconstruct cerebellar folding. Histological data, even at low scanning resolution, can provide sufficient information to distinguish folia as well as cerebellar cortical layers. However, data is only 2D, and sectioning often induces non-trivial deformations which make it challenging to create full 3D reconstructions (see, however, the work of *Sereno et al., 2020*, and *Zheng et al., 2023*). Using histological data, *Ashwell, 2020*, was able to estimate the surface area in 90 species of marsupials and 57 species of eutherian mammals, but did not study folia, of which the measurement remains challenging.

In our analyses we manually segmented the surface of the cerebellum and developed a method to automatically detect the boundary between the molecular and granular layers, allowing us to estimate the median thickness of the molecular layer. We also developed a method to detect individual folia, and to measure their median width and perimeter. Buckling models of brain folding suggest an inverse relationship between folding frequency and cortical thickness, with thicker cortices leading to lower frequency folding (*Toro and Burnod, 2005*). This has been verified in the mammalian cerebrum (*Mota and Herculano-Houzel, 2015*), and we aimed at studying it in the cerebellum. The analysis of phylogenetic data is challenging, because different species share a varying degree of genetic relationship. We used phylogenetic comparative methods to condition our analyses on the phylogenetic tree and tested different evolutionary scenarios, which allowed us to study the evolution of the cerebellum and its relationship with the cerebrum and with body size.

Our results showed two very different groups of phenotypes: those like brain size, that changed over several orders of magnitude, and those like folial width which were much more conserved. We confirm the strong allometry between the size of the cerebellum and the cerebrum across mammals (*Barton and Harvey, 2000*; *Barton, 2002*; *Whiting and Barton, 2003*; *Herculano-Houzel, 2010*; *Barton, 2012*; *Smaers et al., 2018*; *Ashwell, 2020*; *Magielse et al., 2023*) and show similarly strong relationships for the width of cerebellar folds and the thickness of the molecular layer, although with a much narrower range of variation. This deeply conserved pattern suggests the presence of a common mechanism underlying the development of the cerebellum and cerebellar folding across mammals. All our data and code are openly available online.

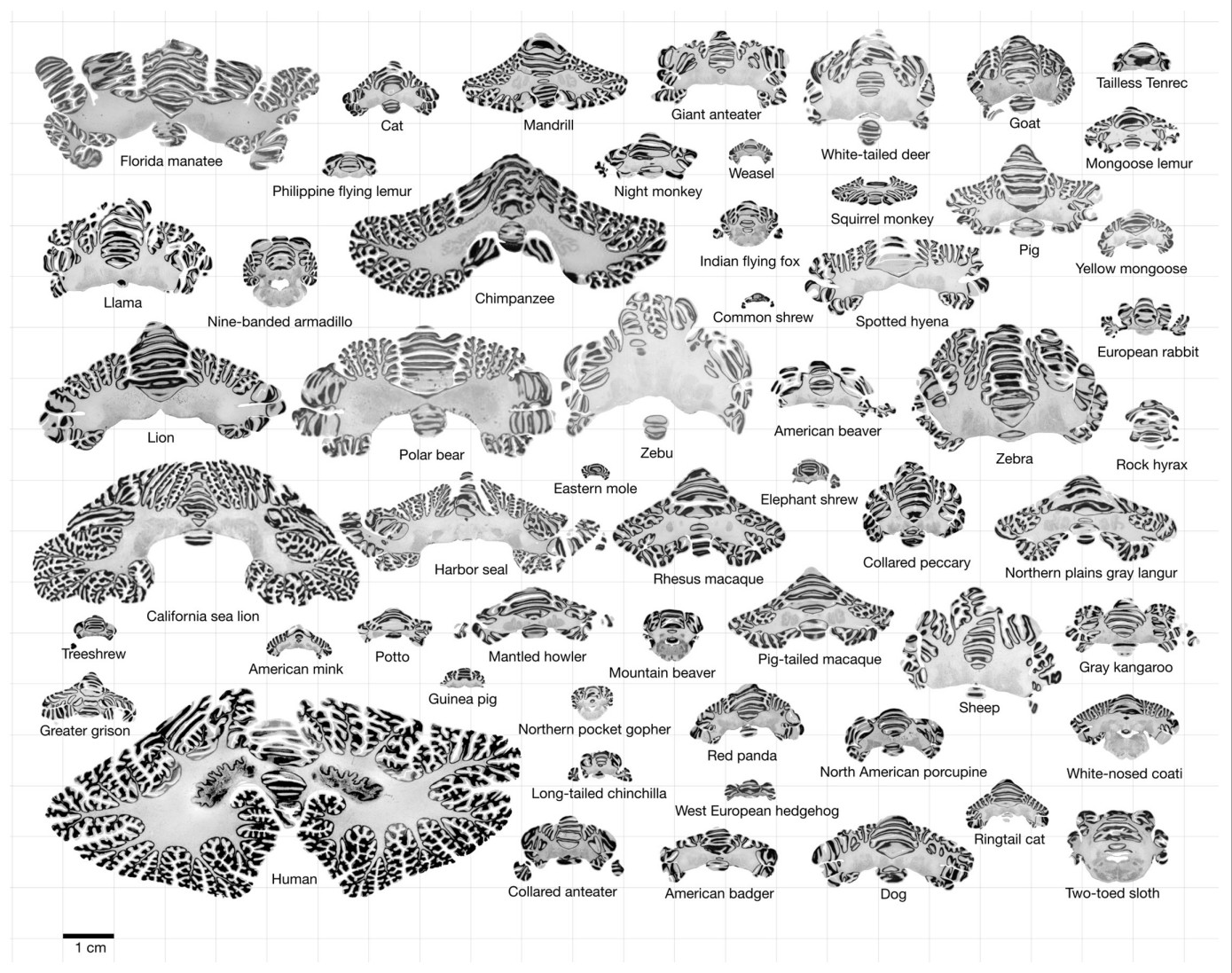

**Figure 1.** Data. All coronal cerebellar mid-sections for the different species analysed, at the same scale. This dataset is available online for interactive visualisation and annotation: https://microdraw.pasteur.fr/project/brainmuseum-cb. Image available at https://doi.org/10.5281/zenodo.8020178.

## Materials and methods

### Histological data

Most of the data used in this study comes from the Brain Museum website (https://brainmuseum.org) – the Comparative Mammalian Brain Collection initiated by Wally Welker, John Irwin Johnson, and Adrianne Noe from the University of Wisconsin, Michigan State University, and the National Museum of Health and Medicine. This is the same eutherian mammal data used by *Ashwell, 2020*, through the Neuroscience Library website (https://neurosciencelibrary.org, offline since June 2022). The human brain data comes from the BigBrain Project (https://bigbrainproject.org) led by teams from Forschungszentrum Jülich and McGill University (*Amunts et al., 2013*). The rhesus macaque data is part of the BrainMaps project (http://brainmaps.org) by Edward G Jones at UC Davis (*Figure 1* and *Supplementary file 1* for a detailed list).

We used our Web tool MicroDraw to visualise and segment the histological data online (https://microdraw.pasteur.fr). We downloaded the Brain Museum data, and used the vips library (https://www.libvips.org) to convert it to deep zoom image format – the multi-scale image format required by MicroDraw. The data from the BigBrain and BrainMaps projects is already encoded in multi-scale

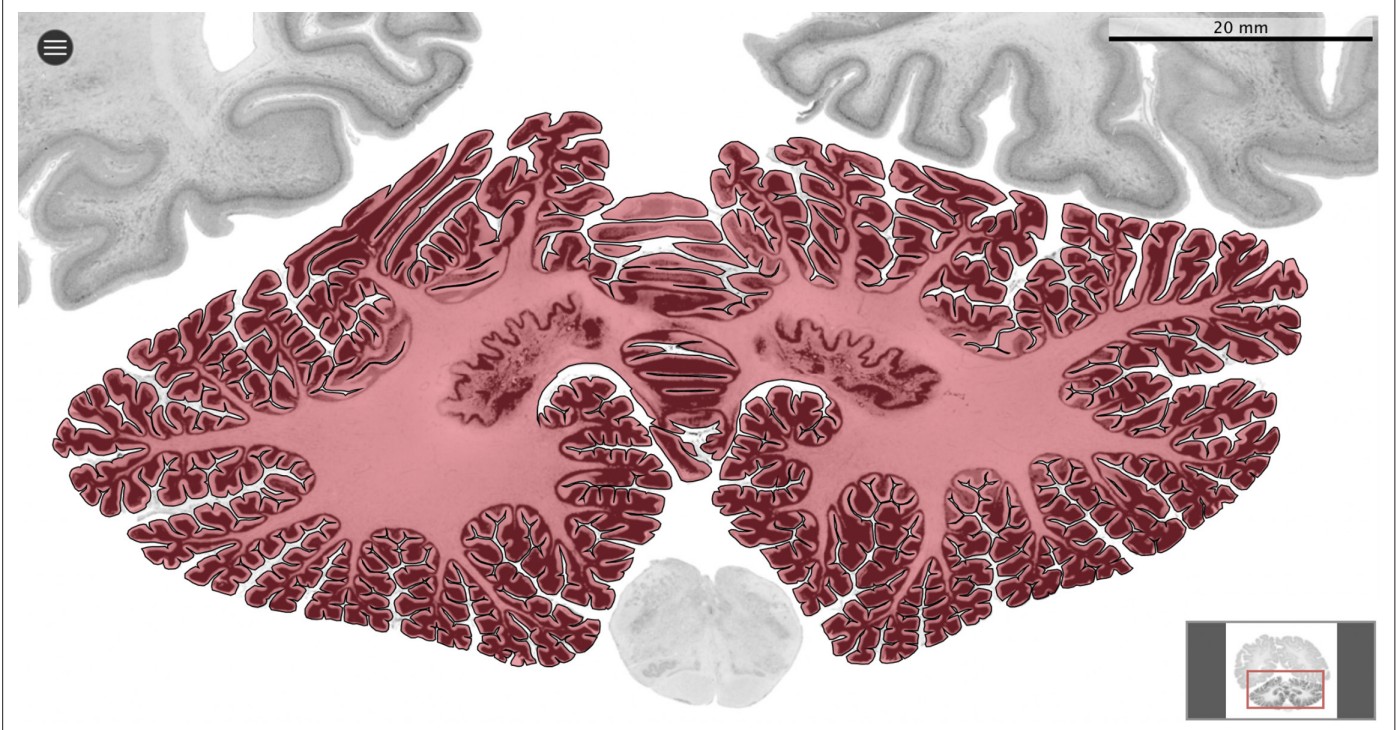

**Figure 2.** Segmentation. All datasets were indexed in our collaborative Web application MicroDraw (https://microdraw.pasteur.fr) to interactively view and annotate the sections. The contour of the cerebellum was drawn manually using MicroDraw (black contour). The example image shows the human cerebellum from the BigBrain 20 µm dataset (**Amunts et al., 2013**).

format, and we only wrote translation functions to allow access through MicroDraw. The Brain Museum Cerebellum project can be accessed at https://microdraw.pasteur.fr/project/brainmuseum-cb.

The resolution of the histological images in the Brain Museum was very variable. We considered that image resolution was in all cases sufficient to estimate the length and area of the cerebellum sections, however, we excluded some species from our estimation of molecular layer thickness when resolution was deemed insufficient. For this, we plotted the total number of pixels of the image versus cerebellum size, and excluded species where the density of pixels per $mm^2$ was lower than 3.5 – a threshold determined by visual inspection. The datasets from the BrainMaps (rhesus macaque), and the BigBrain project (human) were all high resolution (1–20 µm).

## Segmentation

We used MicroDraw to draw the scale bar, and the contour of the coronal mid-section of the cerebellum (**Figure 1**), and the cerebrum (for each numbered series of coronal sections spanning the length of the structure, the section with the median number for that particular structure). In the case of the human brain from the BigBrain project, and the rhesus macaque from the BrainMaps project, where no scale bar is present directly in the sections, we obtained scale information from their websites (**Figure 2**).

## Measurement of section area and length of the cerebral and cerebellar cortex

*Table 1* summarises the variables used in our analyses and their sources. We used the Python package microdraw.py (https://github.com/neuroanatomy/microdraw.py; **Heuer et al., 2023**) to query Micro-Draw's API and programmatically download all vectorial segmentations and images. We computed the length of the cerebral and cerebellar segmentation contours, and used the scale bar information to convert the data to millimetres. We used an artificial calibration image to ensure that our algorithms produce the correct measurements. Measurements were not modified to account for shrinkage, and reflect the raw data obtained from the images (see **Ashwell, 2020**, for a discussion of shrinkage in the Brain Museum collection). All measurements were $\log_{10}$ converted to facilitate their comparison.

**Table 1.** Definition of the variables used in the current study and their sources.

| Variable name | Definition | Source |
|---|---|---|
| Cerebellar section area | Area of the cerebellar mid-section (mm²) | Measured |
| Cerebellar section length | Perimeter around the cerebellar pial surface of mid-section (mm) | Measured |
| Folial width (cerebellum) | Euclidean distance between the two flanking sulci of a folium, averaged for all folia in the mid-section (mm; *Figure 3b*) | Measured |
| Folial perimeter (cerebellum) | Perimeter along the cerebellar pial surface between the two flanking sulci of a folium, averaged for all folia in the mid section (mm; *Figure 3c*) | Measured |
| Thickness (cerebellum) | Thickness of the molecular layer for the cerebellar mid section, averaged from the lengths profile lines that bisect the molecular layer from its border with the pial surface to its border with the granular surface (mm; *Figure 4*) | Measured |
| Cerebral section area | Area of the cerebral mid-section (mm²) | Measured |
| Cerebral section length | Perimeter around the cerebral pial surface of mid-section (mm) | Measured |
| Brain weight | Weight of the whole brain, including cerebrum and cerebellum (g) | *Ballarin et al., 2016*; *Burger et al., 2019*; *Smaers et al., 2021* |
| Body weight | Weight of the whole body (g) | *Ballarin et al., 2016*; *Burger et al., 2019*; *Smaers et al., 2021* |
| Cerebellar volume | Volume of the entire cerebellum, estimated from 2D histological sections or MRI (mm³) | *Smaers et al., 2018* |
| Cerebral volume | Volume of the entire cerebrum, estimated from 2D histological sections or MRI (mm³) | *Smaers et al., 2018* |
| Sultan's folial width | Length of a cerebellar fold in a cross-section orthogonal to the fold axis: each fold may contain several individual folia | *Sultan and Braitenberg, 1993* |

## Measurement of folial width and perimeter

We defined two measures of folding – folding width and folding perimeter – which are closely related to folding frequency and amplitude (*Figure 3a–c*). As shown in *Figure 3d*, they allow us to differentiate among contours with the same gyrification index – a method introduced by *Zilles et al., 1988*, which has been widely used to study the degree of cerebellar folding.

To compute folding width and folding perimeter we resampled the manual segmentations to have a homogeneous density of vertices and automatically detected sulci (*Figure 3a*). At each vertex we measured the mean curvature of the segmentation path. We smoothed the mean curvature measurements at 10 different scales, which produced for each vertex a 10D signature. We used a random forest algorithm to automatically distinguish three classes of vertices: gyri, sulci, and walls. We measured the width of each folium as the Euclidean distance between its two flanking sulci (*Figure 3b*). We measured the perimeter of each folium as the distance along the cerebellar surface between its two flanking sulci (*Figure 3c*). For each species we used the median width and the median perimeter across all folia, to make our estimations robust to outliers. The code for computing folial width and perimeter is provided in the accompanying GitHub repository.

## Measurement of molecular layer thickness

We estimated the thickness of the molecular layer automatically (*Figure 4*). We processed the histological images to convert them to grey levels, denoise them, and equalise their histograms, using functions from the Scikit-image package (*van der Walt et al., 2014*). Starting from the manual segmentation (blue line in *Figure 4a*), we created a binary mask and masked out the non-cerebellar

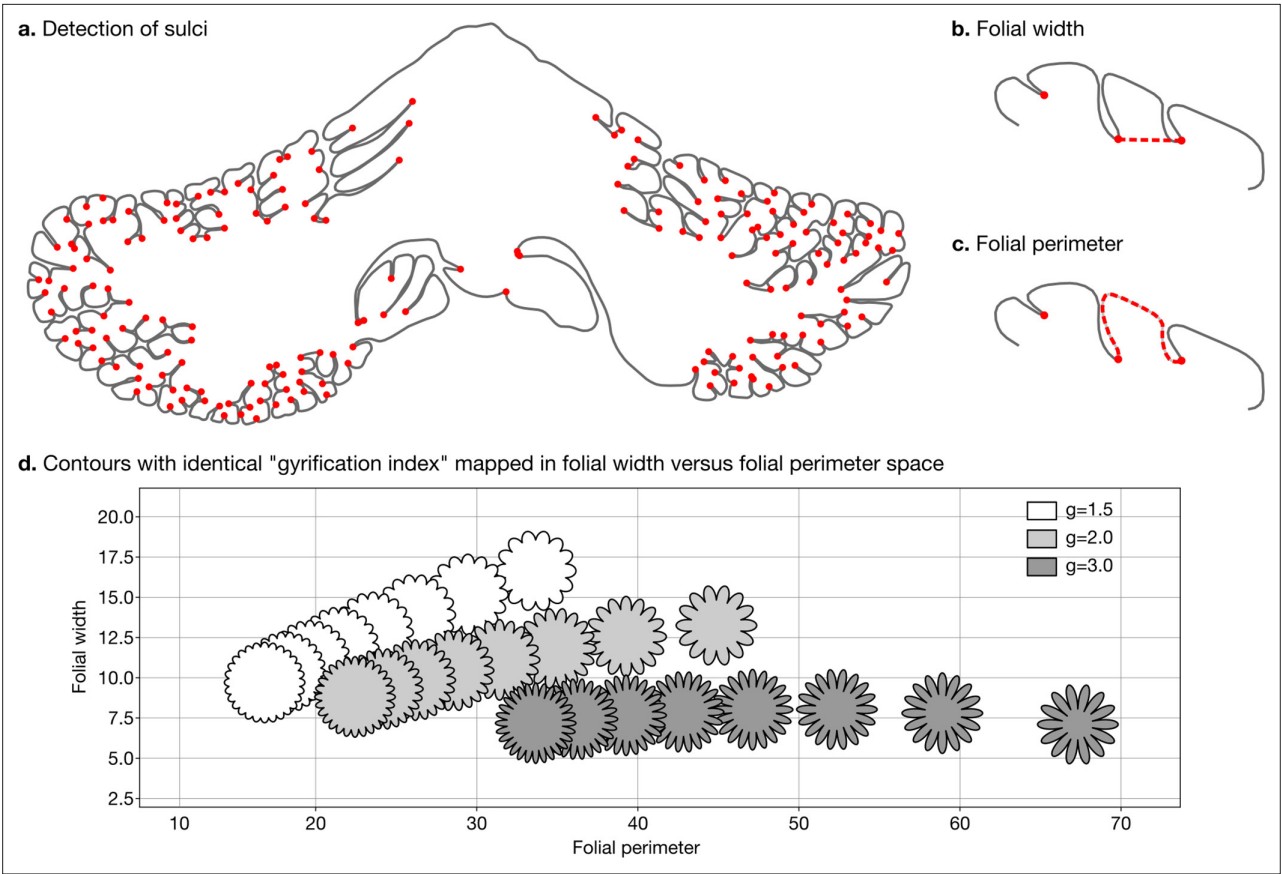

**Figure 3.** Detection of sulci and measurement of folial width and perimeter. Sulci were automatically detected using mean curvature filtered at different scales (**a**). From this, we computed folial width (**b**) and folial perimeter (**c**). (**d**) Folial width and perimeter allow to distinguish folding patterns with the same gyrification index. The three rows of contours have the same gyrification index (top $g$=1.5, middle $g$=2.0, bottom $g$=3.0), however, the gyrification index does not allow distinguishing between contours with many shallow folds and those with few deep ones.

regions (e.g., the neighbouring cerebrum). The molecular layer of the cerebellum appears as a light band followed by the granular layer which appears as a dark band. We applied a Laplacian smoothing to the masked image, which created a gradual change in grey level going from white close to the surface of the cerebellum to black towards the granular layer. We computed the gradient of the image, which produced a vector field from the outside to the inside. We integrated this vector field to produce a series of lines, one for every vertex in the manual cerebellar segmentation. The vector field was integrated only if the grey values decreased, and if a minimum was reached, the vector field was continued linearly. At the end of this procedure, all lines had the same length, computed so as to cover the whole range from the surface of the cerebellum to the granular layer. The grey levels in the original image were sampled along each scan line, producing a grey level profile. The grey level profiles were derived, and a peak detection function was used to determine the point of maximum white/grey contrast (*Figure 4c*, where the red dot shows the detected boundary), indicating the boundary between the molecular and granular layers. *Figure 4b* shows the part of the profile lines starting at the cerebellar surface and ending at the detected boundary. For each scan line the corresponding thickness of the molecular layer was defined as the length from the manual cerebellar segmentation until the maximum contrast point (red point). Finally, for each species, a single thickness value was computed as the median of the thickness measured for each profile line, to make our estimation robust to outliers. See the accompanying source code for further implementation details.

## Phylogenetic comparative analyses

We obtained the phylogenetic tree for our 56 species from the TimeTree website (*Figure 5a*, https://www.timetree.org, *Kumar et al., 2017*). The analysis of neuroanatomical variation across species

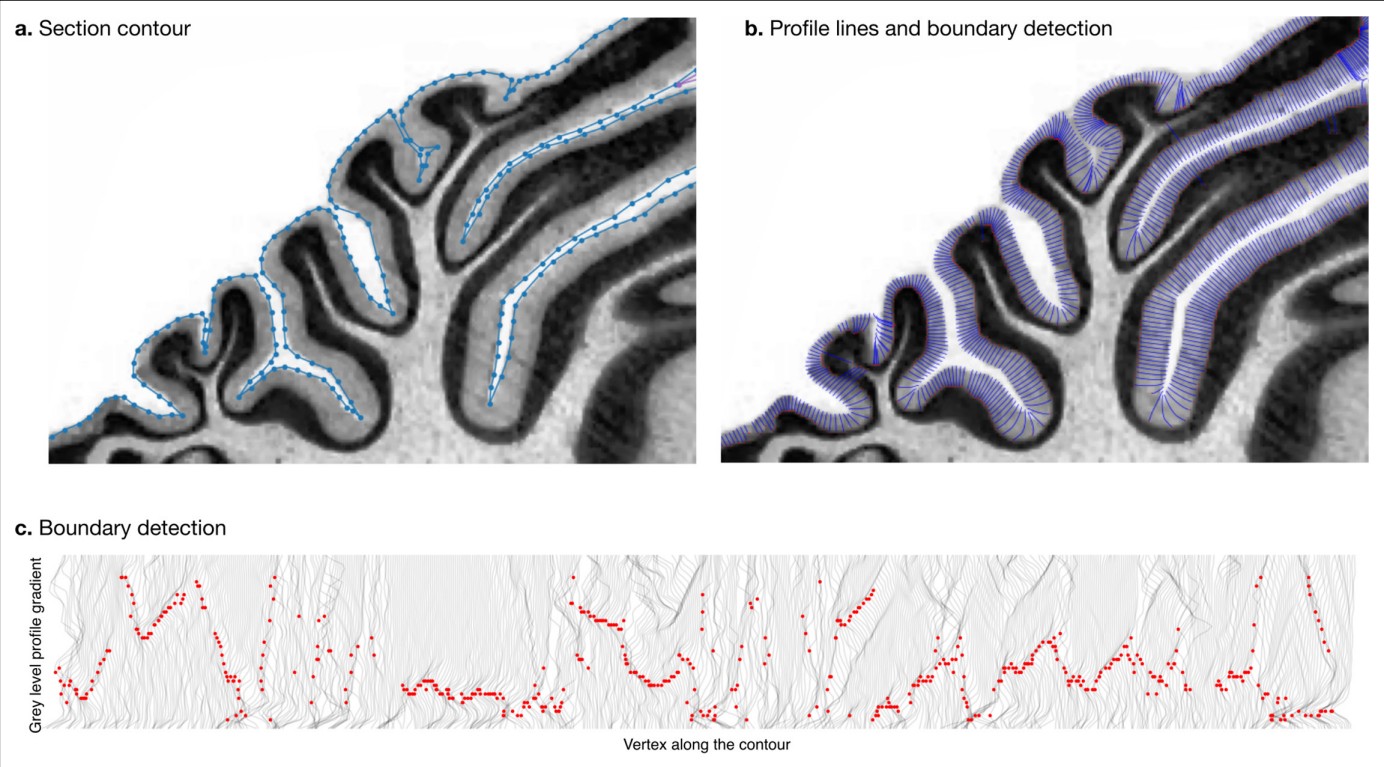

**a.** Section contour

**b.** Profile lines and boundary detection

**c.** Boundary detection

*Grey level profile gradient*

*Vertex along the contour*

**Figure 4.** Measurement of molecular layer thickness. The thickness of the molecular layer was measured automatically from the surface segmentation. (**a**) Zoom into a manually annotated surface contour. (**b**) Automatically computed profile lines. (**c**) Grey level profile gradients and border detection for the whole slice. Red dots indicate detected borders (maximum gradient).

and through evolution requires taking these phylogenetic relationships into account, and naively comparing different clades may lead to misinterpretations because the measurements are not independent. Phenotypic differences between clades can reflect different adaptations, but they can also reflect their different phylogenetic relationships.

Consider for example a phenotype varying randomly. Species descending from a recent common ancestor would be expected to be more similar simply because not enough time has passed for them to drift apart (*Figure 5b*, adapted from *Felsenstein, 1985*). This would not be the case if the split from their common ancestor had been more distant in time, and their phenotype would be much more different (*Figure 5c*). A more rigorous approach takes into account the whole hierarchy of relationships, as described by a phylogenetic tree, which is used to condition the structure of variation across species.

In addition to the tree structure, the variation of phenotypes along its branches can be modelled by different evolutionary processes, which can be explicitly formulated, fitted to the data, and compared. We tested three different models. First, the randomly varying phenotype that we mentioned could be modelled as a Brownian motion (BM) process. BM can be used to model fluctuating selection and genetic drift, for example. Once an ancestral species splits into two new species, their phenotypes will become progressively decorrelated: $dX(t) = \sigma dB(t)$, where $dX(t)$ indicates an infinitesimal change in the trait $X$ at time $t$, $\sigma$ is the magnitude of change in $X(t)$, and $dB(t)$ is a scatter generator (*Cavalli-Sforza and Edwards, 1967*). Second, the Ornstein-Uhlenbeck (OU) model extends the BM model by including the idea of an optimal trait value: $dX(t) = \alpha\left(\theta - X(t)\right)dt + \sigma dB(t)$, where θ is the optimum trait value, and α is the strength of attraction to θ. Traits will vary with a random component, as in the BM model, but will also be attracted to the optimum value, which has been used to model stabilising selection (*Felsenstein, 1988*; *Hansen and Martins, 1996*). Finally, the early burst (EB) model (*Harmon et al., 2010*), also known as the accelerating-decelerating or ACDC model, (*Blomberg et al., 2003*) is an alternative extension of the BM model where the magnitude of random change is modulated by an exponential function: $dX(t) = \sigma e^{rt/2}dB(t)$, where r is the strength of exponential change. EB models

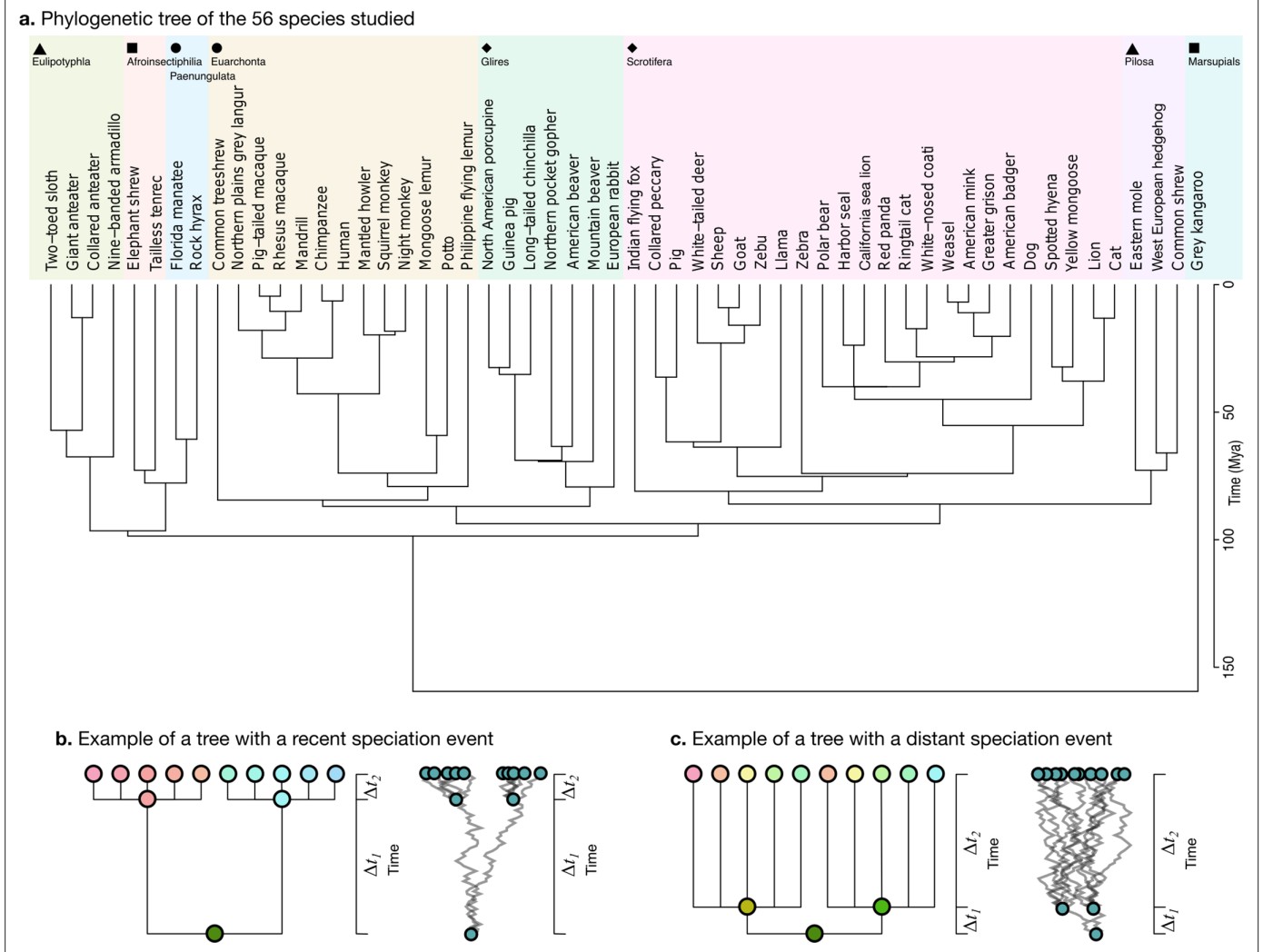

**Figure 5.** Phylogenetic tree. (**a**) The phylogenetic tree for 56 species used in our study was downloaded from the TimeTree website (http://www.timetree.org, **Kumar et al., 2017**) and coloured in eight groups based on hierarchical clustering of the tree. (**b**) Not accounting for phylogenetic information can lead to misinterpretations of cross-species relationships. Consider the case of eight species descending from two recent ancestors as in the tree ($\Delta t_2 \ll \Delta t_1$, adapted from **Felsenstein, 1985**). Even if their phenotypes evolved randomly (Brownian motion [BM] process in the phenogram to the right), they will seem to be more similar within each group. (**c**) If the time of split from their common ancestor were more distant ($\Delta t_2 \gg \Delta t_1$), this difference would not exist. Real phylogenetic trees embed a complex hierarchical structure of such relationships.

a rapid diversification of species in the early stages of their evolution, followed by a slowdown in the rate of diversification (**Blomberg et al., 2003**; **Harmon et al., 2010**).

Multivariate models were fitted using the mvMORPH package (**Clavel et al., 2015**). We selected the best fitting model using the Akaike information criterion (AIC), corrected for small sample sizes (AICc). AIC takes into account the number of parameters $p$ in the model: $AIC = -2log\left(likelihood\right) + 2p$. This approximation is insufficient when the sample size is small, in which case an additional correction is required, leading to the corrected AIC: $AICc = AIC + \left(2p^2 + 2p\right) / \left(n - p - 1\right)$, where $n$ is the sample size.

The analysis of phylogenetic data is challenging especially when the number of species is reduced, the number of traits studied is large and the models to fit are complex. **Cooper et al., 2016**, have warned in particular about fitting OU models using likelihood-based methods, and the possibility of incorrectly selecting them even when data was generated from a BM process. These difficulties can be addressed, for example, through the use of AICc instead of likelihood for model selection, as well as the use of penalised likelihood frameworks, which is the approach implemented in mvMORPH (**Clavel**

*et al., 2015*; *Clavel et al., 2019*). To better understand our ability to decide among different models, we estimated our statistical power through simulations. We ran 1000 simulations where each time we generated nine correlated traits following a BM process, and counted the number of times the BM model was selected over the OU model based on their AICc.

## Correlation structure, bivariate, and multivariate allometry

We aimed at describing the main patterns of variation among our phenotypes by looking at their matrix of correlations, partial correlations, principal components, and allometry. Our measurements were not independent because species are linked by different phylogenetic relationships. Our analyses used the phylogenetic tree to model covariance structure, as well as the model selected using the methods described in the previous subsection.

All our phenotypes were strongly correlated. We used partial correlations to better understand pairwise relationships. The partial correlation between two vectors of measurements *a* and *b* is the correlation between their residuals after the influence of all other measurements has been covaried out. Even if the correlation between *a* and *b* is strong and positive, their partial correlation could be 0 or even negative. Consider, for example, three vectors of measurements *a*, *b*, *c*, which result from the combination of uncorrelated random vectors *x*, *y*, *z*. Suppose that *a*=0.5*x*+0.2*y*+0.1*z*, *b*=0.5*x* - 0.2*y*+0.1*z*, and *c*=*x*. The measurements *a* and *b* will be positively correlated because of the effect of *x* and *z*. However, if we compute the residuals of *a* and *b* after covarying the effect of *c* (i.e., *x*), their partial correlation will be negative because of the opposite effect of *y* on *a* and *b*. The statistical significance of each partial correlation being different from 0 was estimated using the edge exclusion test introduced by *Whittaker, 1990*. We evaluated our statistical power to detect partial correlations of different strengths using simulations. Code is included in the accompanying source code along with a Jupyter notebook providing an executable version of our partial correlations example together with further details on our power analysis. A non-executable version of this notebook is provided as *Supplementary file 2*.

The main patterns of correlation among phenotypes, that is, the way in which different groups of phenotypes vary together, can be further studied using principal component analysis (PCA). The first principal component (PC1) is a vector of loadings showing how much each variable contributes to the main pattern of variation. The loadings of the second principal component (PC2) show the contribution of each variable to the second most important pattern of variation, etc. Each principal component captures a certain amount of variance, which indicates the importance of the pattern of variation it represents. By construction, the different principal components are orthogonal (another way of understanding PCA is as the rigid rotation of the data that will produce the best alignment with a set of orthogonal axes). Since species are non-independent data points due to evolutionary relatedness, it is important to take phylogenetic tree structure into account. For example, in the case of traits generated by a BM process, the main axis of variation could be due to traits being different because of biological reasons, or just because the species where they were measured had diverged a long time ago as in *Figure 5c* (*Felsenstein, 1985*; *Revell, 2012*; *Clavel et al., 2015*).

Finally, we also use allometry to study the relationships between measurements in organisms of different shape and size. Consider, for example, the case of isometry: when a series of objects have identical shapes but different sizes. Their surface area will increase proportionally to the second power of their length, and volume will increase as the third power of their length. A measurement is said to be hypo-allometric or to have negative allometry relative to another if it changes less than what would be expected by isometry. This is the case of brain volume, which increases proportionally less than body volume across species. A measurement is said to be hyper-allometric or to have positive allometry relative to another if its changes are larger than what would be expected by isometry. For example, cortical surface area increases disproportionately when brain volume increases: in the case of isometry, we would expect the surface area of an object to increase proportionally to the ⅔ power of its volume. In mammals, however, the increase is almost linear (which is explained by the presence of brain folding, see *Toro et al., 2008*). The allometric scaling between two measurements is often estimated by considering the slope of their linear regression in a log-log scale. More accurately, orthogonal regression can be used, which distributes errors across both variables. The orthogonal regression approach can be also easily extended to the case of multivariate allometry: the loadings of the first principal component of the log-converted measurements allow us to estimate the allometric

scaling of all variables at the same time (*Jolicoeur, 1963*). Consider the case of volumetric measurements of different brain regions (log-converted). The loadings of PC1 will indicate the contribution of the corresponding brain region to the main pattern of variation, and the scaling between any pair of regions can be obtained as the ratio between their corresponding loadings (see, for example, *Toro et al., 2009*).

In what follows we report allometric slopes obtained using linear regression and orthogonal regression for selected pairs of variables, as well as the slopes obtained using multivariate allometry (i.e., ratios of loadings from their phylogeny-controlled PC1).

### Estimation of ancestral phenotypes

Fitting of evolutionary models allows us to obtain estimation of the ancestral states of the phenotypes under study. Our phenotypes showed a strong correlation pattern. Instead of estimating ancestral states for each of them independently, we estimated the ancestral state of the first two principal components for all nine phenotypes, and also for a reduced set of phenotypes including only neuroanatomical measurements.

All the code necessary to reproduce our analyses is available at https://github.com/neuroanatomy/comp-cb-folding (copy archived at *Heuer et al., 2023*).

## Results

The presentation of our results is structured as follows: (1) we validate our measurements by comparing them with the literature, (2) we show the results of the fit of the three different evolutionary models to our data, and decide on the one which will be used in the following analyses to constrain the variance structure, (3) we describe the structure of correlations among our phenotypes and present their main multivariate patterns of variation, two different sets of phenotypes appeared, a first set showing large variation across several orders of magnitude, and a second set which is much more conserved in comparison. (4) We present bivariate allometry results to illustrate these relationships, (5) finally we estimate the ancestral states of the two main patterns of variation in the data, which provides us with an idea of their evolutionary history.

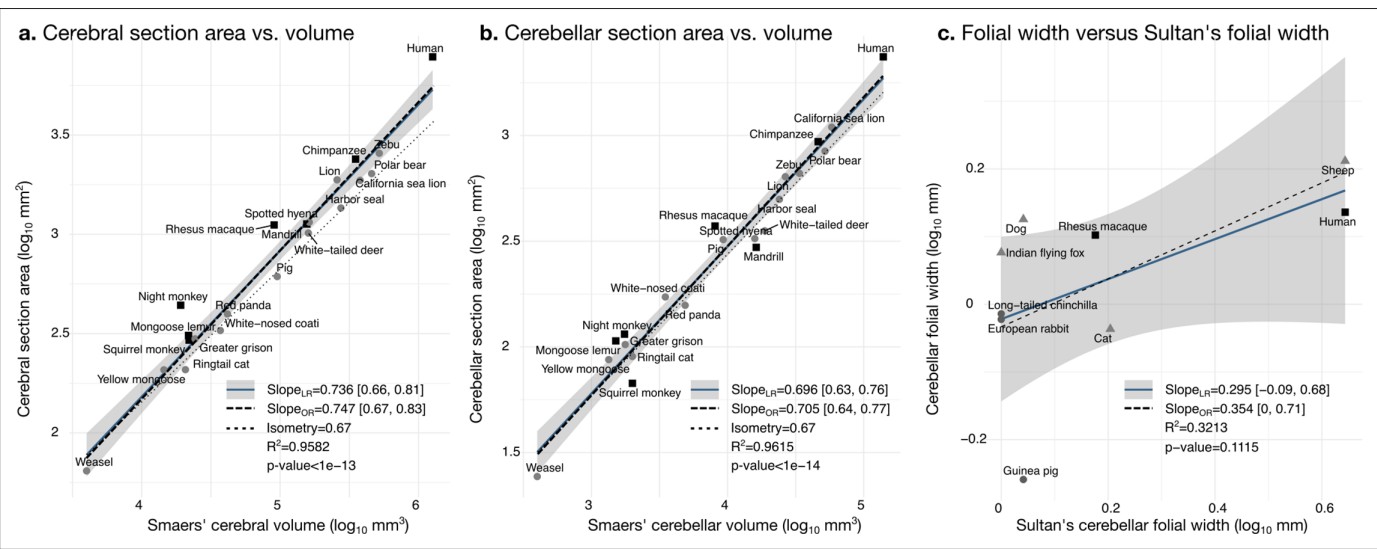

**Figure 6.** Validation of neuroanatomical measurements. Comparison of our cortical section area measurement (**a**) and cerebellar section area (**b**) with volume measurements from *Smaers et al., 2018*. (**c**) Comparison of our folial width measurement with folial width from *Sultan and Braitenberg, 1993*. The folial width measurement reported by Sultan and Braitenberg, which may include several folia, do not correlate significantly with our measurements (two-tailed p=0.112). LR: linear regression with 95% confidence interval in grey. OR: orthogonal regression.

The online version of this article includes the following figure supplement(s) for figure 6:

**Figure supplement 1.** Validation of neuroanatomical measurements: correlation with measurements in *Ashwell, 2020*.

## Validation of measurements

### Cerebellar and cerebral section area

Our measurements were a good proxy for total volume and correlated well with those reported in the literature. We segmented the coronal mid-section of the cerebellum and the cerebrum. These section areas should scale as the 2/3 power of the volume. We confirmed that this was the case by comparing our measurements with the volume measurements reported by *Smaers et al., 2018*: the correlation captured ~96% of the variance (*Figure 6a and b*). *Smaers et al., 2018*, report separate values for the medial cerebellum (the vermal region) (based on measurements from *MacLeod et al., 2003*; *Maseko et al., 2012*; *Smaers et al., 2011*). Our cerebellum section area measurement also correlated strongly with their medial cerebellum volume measurement, a fact that could be eventually used in the future for imputing missing data, although we do not use medial cerebellar measurements in our analyses. See *Figure 6—figure supplement 1* for correlation with measurements in *Ashwell, 2020*.

### Folding frequency

There are no comparative analyses of the frequency of cerebellar folding, to our knowledge. The closest is the measurements reported by *Sultan and Braitenberg, 1993*, measuring the length of structures which may include several individual folia. Our measurements of cerebellar folial width and folial perimeter, however, do not correlate significantly with Sultan and Braitenberg's measurements (*Figure 6c*). *Ashwell, 2020*, studied cerebellar folding through a foliation index, defined as the ratio of the cerebellar pial surface over the external cerebellar surface. This method, however, does not provide information about folding frequency, and would be unable to distinguish a cerebellum with many shallow folds from one with a few deep ones, which could both produce a similar foliation index (e.g., *Figure 3d*).

### Molecular layer thickness

Although several previous works report volume of the molecular layer, we were unable to find data on molecular layer thickness across species. *Zheng et al., 2023*, reported 0.32 mm (±0.08) for humans, and our estimation was of 0.29 mm, which is well within the confidence interval.

## Evolutionary model selection

### The evolution of cerebellar and cerebral neuroanatomy follows a stabilising selection process

We compared three different models of the evolution of cerebellar and cerebral neuroanatomical measurements: BM, Ornstein-Uhlenbeck (OU), and early burst (EB). Comparing the goodness of fit of the three models, we observed substantial evidence in favour of the OU model (*Table 2*, smaller values indicate a better fit). The second most likely model was EB, and finally BM.

The $\theta$ parameters in the selected OU model (optimal trait values) were close to the mean of our sample and the $\alpha$ values were relatively strong, suggesting a process that dampens extreme phenotypic values. The $\alpha$ values can be transformed into a 'halflife' which can be interpreted as the time it would take the evolutionary process to bring a trait to half of the optimal trait value. The average halflife = $\log(2)/\alpha$ was 20 million years.

To address the concern of a bias in the selection of OU models (*Cooper et al., 2016*), we used simulations to study our ability to correctly detect a BM process (the most parsimonious of our three models) through simulation, using our phylogenetic tree and nine randomly generated correlated

**Table 2.** Ranking of phylogenetic comparative models.
Results based on the Akaike information criterion corrected for small sample sizes (AICc). The best fit to the data (smallest AICc value) was obtained for the Ornstein-Uhlenbeck model.

| Ranking | Model | AICc | Log-likelihood |
|---|---|---|---|
| 1 | Ornstein-Uhlenbeck (OU) | −360.64 | 252.48 |
| 2 | Early burst (EB) | −254.03 | 188.89 |
| 3 | Brownian motion (BM) | −227.73 | 174.62 |

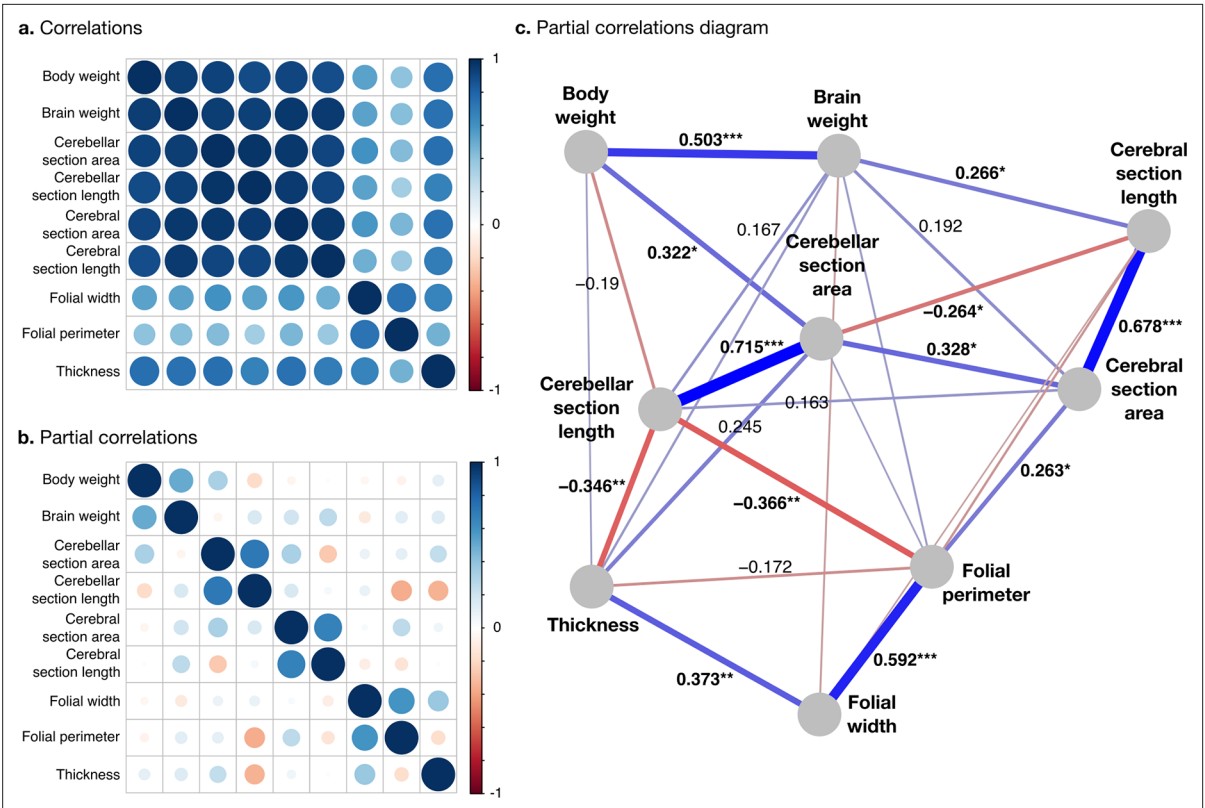

**Figure 7.** Correlation structure. (**a**) Correlation matrix among all phenotypes (all values log₁₀ transformed). (**b**) Partial correlation matrix. (**c**) Graph representation of the strongest positive (red) and negative (blue) partial correlations. All correlations are conditional to phylogenetic tree data. Partial correlation significance is indicated by asterisks, *** for p<0.001, ** for p<0.01, * for p<0.05. Partial correlations without asterisks are not significant.

phenotypes. In 1000 simulations, the BM model was correctly selected over the OU model 99.3% of the times (see Supplemental Methods for details). The analyses of relationships among phenotypes that follow use the OU model to control for phylogenetic structure (otherwise, the observation would not be evolutionarily independent).

## Correlation structure

### Anatomical phenotypes segregate into 'diverse' and 'stable' groups

All measurements were strongly positively correlated (*Figure 7a*). To better understand the global structure of these correlations, we computed partial correlations, which are the correlations between pairs of variables after controlling for the effect of all other variables. Two groups of phenotypes appeared: First, a group of 'diverse' phenotypes that varied widely across species, including body weight, brain weight, cerebellar, and cerebral section area and length, characterised by strong positive partial correlations in the range from 0.3 to 0.7. Cerebellar section area, for example, varies over ~2.5 orders of magnitude, and body weight over 10.8 orders of magnitude. Second, a group of 'stable' phenotypes that showed much less variation in comparison, including folial width, folial period, and thickness of the molecular layer, with positive partial correlations in the range from 0.4 to 0.6. There was only a <4-fold variation in these phenotypes across species (~0.5 order of magnitude).

Phenotypes within the diverse and stable groups showed significant positive within-group partial correlations. Within the diverse group, the most important positive partial correlation between cerebellum and cerebrum was between cerebral section area (a 2D section of cerebral volume) and cerebellar section area (a 2D section of cerebellar volume). The diverse and the stable groups were linked through significant negative correlations, which were the largest we observed. They linked (1) cerebellar section length and folial perimeter and (2) cerebellar section length and thickness of the molecular layer (p-values <10⁻²), indicating that larger cerebella tend to have relatively smaller and thinner folia than what could be expected for their size.

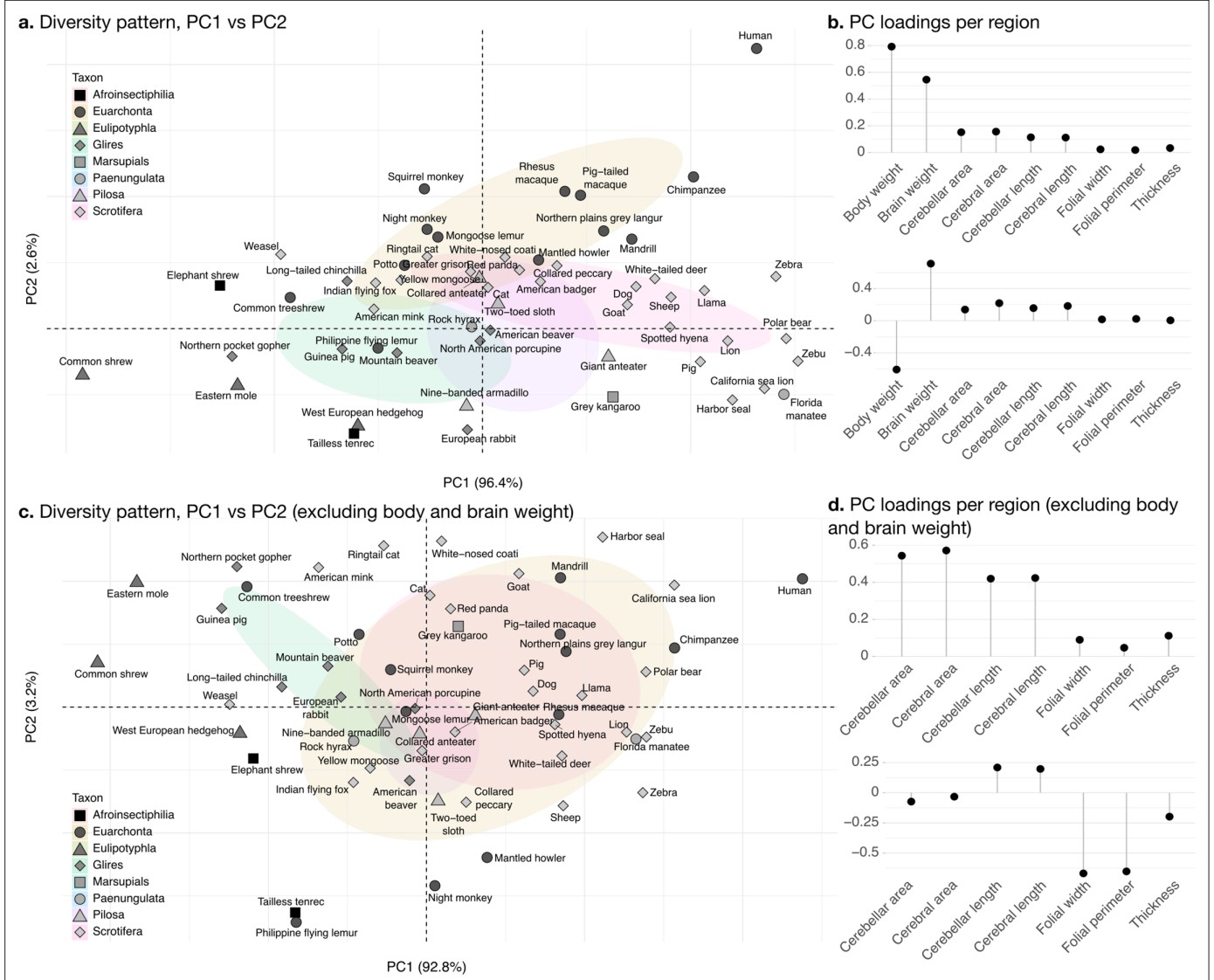

**Figure 8.** Phylogenetic principal component analysis. Primates, and especially humans, show particularly large brains given their body size. (**a**) Pattern of neuroanatomical diversity (PC1 vs. PC2), including body and brain weight. (**b**) Loadings of the PC1 and PC2 axes displayed in panel a. (**c**) Pattern of neuroanatomical diversity (PC1 vs. PC2), excluding body and brain weight. (**d**) Loadings of the PC1 and PC2 axes displayed in panel c.

Our simulations showed that, because of the strength of the allometric pattern in our data, we had excellent statistical power to detect large and medium partial correlations (90–100%) and good statistical power to detect small partial correlations (~78%, see *Supplementary file 2* for details).

## The strongest pattern of variation concerned phenotypes from the diverse group, followed by those in the stable group

We performed a phylogenetic PCA to explore further the main patterns of neuroanatomical variability.

The phylogenetic PCA of all cerebellar and cerebral measurements produced a PC1 which captured the largest part of the variance: 96.4% (*Figure 8b*). PC1 described a strong concerted change in body size and brain size, cerebellar and cerebral section area and length, and loaded weakly on folial width, folial perimeter, and molecular layer thickness. PC1 describes the most important pattern of coordinated allometric variation (*Jolicoeur, 1963*), which will be studied further in the next subsection.

PC2 conveys the main way in which individuals deviate from the dominant allometric pattern. The proportion of variance captured by PC2 was 2.6%. Because of the prominent role of body weight in PC1, PC2 mostly allowed uncoupling body size from changes in brain anatomy (*Figure 8b*). Plotting all

**Table 3.** Multivariate allometry pattern.
The pattern is represented by the loadings of each measurement in the first principal component (PC1, same as *Figure 8b*). Allometric slopes for each pair of variables can be obtained by dividing their loadings. For example, the slope for the cerebellum section length versus cerebellum section area reported in *Figure 9a* is 0.1138/0.1519 ~ 0.749.

| Body weight | Brain weight | Cerebellar section area | Cerebral section area | Cerebellar section length | Cerebral section length | Folial width | Folial period | Thickness |
|---|---|---|---|---|---|---|---|---|
| 0.7923 | 0.5453 | 0.1519 | 0.1563 | 0.1138 | 0.1117 | 0.0237 | 0.0187 | 0.0336 |

species in PC1/PC2 space (*Figure 8a*) showed that humans were here an important exception, with a small body size relative to brain size. To some extent, this was also the case for other primate species in our sample (chimpanzee, rhesus macaque, etc.).

To better understand the patterns of neuroanatomical variation, we performed an additional PCA excluding body and brain weight. This is because the large variation in body weight, in particular, obscured patterns of variation specific to the brain. This time, PC1 captured 92.1% of the variance while PC2 captured 3.0%. The allometric pattern described by PC1 was similar to the previous one, with strong concerted changes in cerebellar and cerebral section area and length, and less strong changes in folial width, folial perimeter, and molecular layer thickness. PC2 was different, and allowed to uncouple changes in folial width and folial perimeter from changes in molecular layer thickness. Here, humans appeared among the species having the highest frequency of cerebellar folia (smaller folial width and perimeter).

## Allometry

Allometry is the study of the relationships between size and shape. As we mentioned earlier, when objects change only in size without changing shape, the length, surface area, and volume of their parts change in a characteristic way: length as the square root of surface area, surface area as the ⅔ power of the volume, etc. The multivariate allometry pattern of all our measurements, given by PC1, is shown in *Table 3* (also shown in *Figure 8b*). Pairwise allometric scaling factors can be obtained from this table as the ratio between the loadings of the respective measurements.

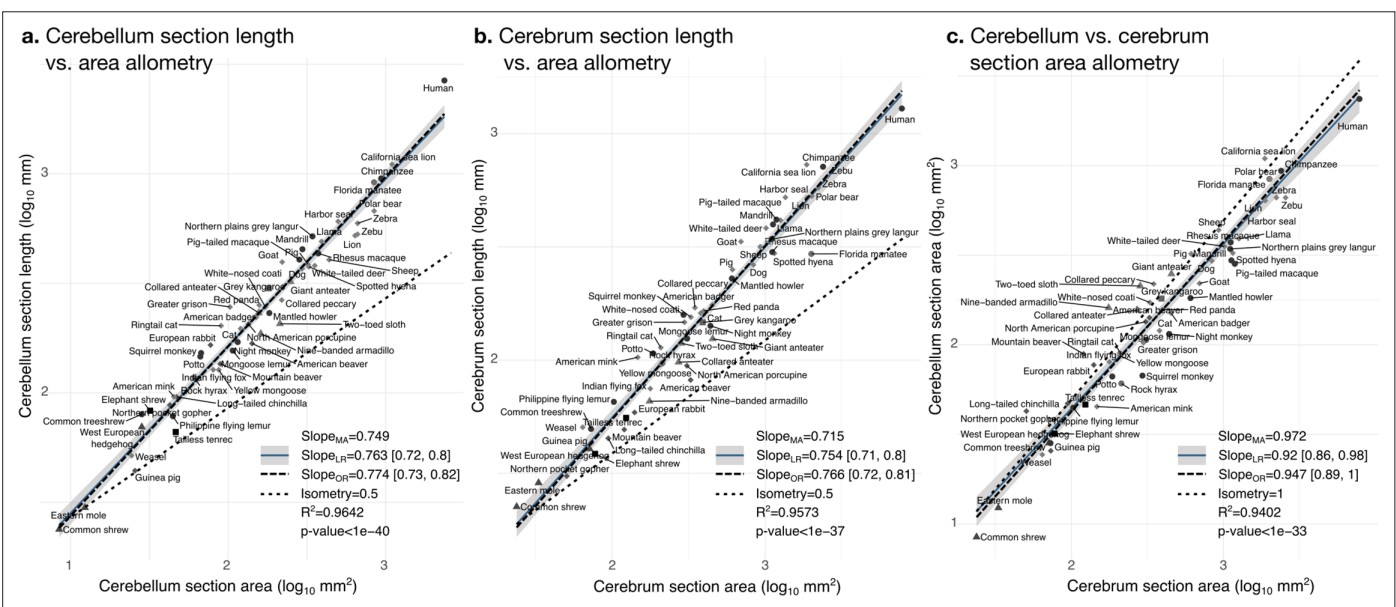

**Figure 9.** Cerebellum and cerebrum folding and allometry. The cerebellar and cerebral cortices are disproportionately larger than their volumes, as shown by their hyper-allometry. The cerebellum is slightly but statistically significantly hypo-allometric compared to the cerebrum. (**a**) Allometry of cerebellum section length vs. cerebellum section area. (**b**) Allometry of cerebrum length vs. cerebrum section area. (**c**) Allometry of cerebellum section area vs. cerebrum section area. MA: multivariate allometry. LR: linear regression with 95% confidence interval in grey. OR: orthogonal regression.

The following analyses use bivariate allometry to focus into particular pairs of phenotypes. First, we focus on those belonging to the 'diverse' group, and then on those belonging to the 'stable' group.

### Cerebellar folding increases with cerebellar size in a similar manner as cerebral folding increases with cerebral size

The bivariate analyses *Figures 9 and 10* zoom into the allometry of cerebellar and cerebral folding. In both cases, our measurement of length – a section of the cerebellar and cerebral surface – correlated strongly with the corresponding measurement of section area – a section of cerebellar and cerebral volume. As we indicated previously, their partial correlations were also strong, suggesting a direct link between both phenotypes. The scaling slopes were clearly higher than 0.5 (the slope of isometric scaling of length versus area), indicating that cerebellar and cerebral cortices are increasingly folded when larger (hyper-allometric scaling). The increase of folding with size followed a similar law for both structures, with a scaling slope of 0.75–0.77 and overlapping 95% confidence intervals. The multivariate allometric slopes were similar to the bivariate slopes: 0.75 for the cerebellum and 0.72 for the cerebrum. In the case of the cerebrum, we confirmed that the Florida manatee was exceptionally unfolded: three times less cortical length than expected. For the cerebellum, we observed that humans had a particularly folded cerebellum: 1.5 times more cerebellar length than expected.

The bivariate plot of cerebellar versus cerebral section area (*Figure 9c*) showed a strong correlation, however, their moderate partial correlation indicates that an important part of this correlation is mediated (including a significant negative partial correlation between cerebellar section area and cerebral section length, *Figure 7c*). The allometric scaling slope suggested that the cerebellum may be slightly proportionally smaller in species with large brains. In the case of isometry, the scaling slope should be 1. The observed scaling slope was slightly smaller than the isometric slope which was nevertheless at the margin of the 95% confidence interval (grey region in the figures), indicating a potential hypo-allometry of the cerebellum.

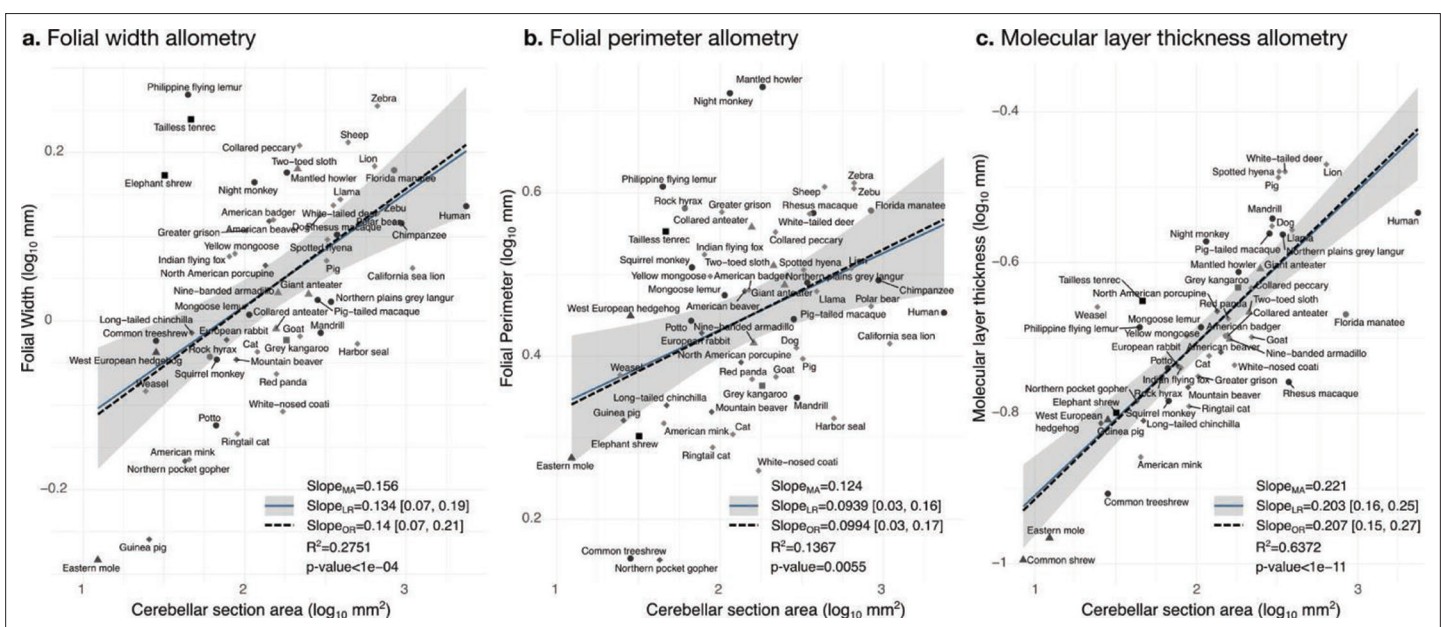

**Figure 10.** Allometry of folial width, perimeter, and molecular layer thickness. The geometry of cerebellar folia and molecular layer thickness were largely conserved when compared with changes in total cerebellar size, as revealed by the small allometric slopes. (**a**) Allometry of folial width vs. cerebellar section area (two-tailed p<1e-5). (**b**) Allometry of folial perimeter vs. cerebellar section area (two-tailed p=0.005). (**c**) Allometry of the thickness of the molecular layer vs. cerebellar section area (two-tailed p<1e-12). MA: multivariate allometry. LR: linear regression with 95% confidence interval in grey. OR: orthogonal regression.

## Folial width, folial perimeter, and the thickness of the molecular layer increase slightly with cerebellar size

Folial width, folial perimeter, and molecular layer thickness are all measurements of length which should also scale as the square root of cerebellar section area in case of isometry. We observed a much smaller scaling factor, indicating that when comparing small and large brains, those measurements increase substantially less than what could be expected from increases in cerebellar size (hypo-allometric scaling). The width and perimeter of cerebellar folia increased only slightly with cerebellar size (*Figure 10*). The scaling was markedly hypo-allometric: 0.10 for folial perimeter and 0.14 for folial width where isometric scaling should be 0.5. The scaling of the molecular layer's thickness was also hypo-allometric, but with a slightly higher scaling slope of 0.2. Folial width and perimeter appeared to be more heterogeneous than molecular layer thickness: linear regression with cerebellar section area captured only 28% of folial width variance, and 14% of folial perimeter variance, while it captured 64% of molecular layer thickness variance (*Figure 10c*).

### Ancestral phenotype estimation

Our ancestral phenotype estimations suggest evolutionary trajectories for the increases in brain size as well as for the decoupling between 'diverse' phenotypes (cerebellar and cerebral section area and length) and 'stable' phenotypes (folial width, folial perimeter, and thickness of the molecular layer). We made estimations of the ancestral states of our phenotypes based on the OU model fit. Our findings are presented for the evolution of PC1 and PC2, which provide a condensed representation of the concerted changes in the nine phenotypes studied. As indicated previously, the large variations in body weight make it the dominating feature in PC1. To zoom into the main patterns of neuroanatomical variation, we also show estimations of ancestral states excluding body and brain weight.

### The increase in body size and brain size have decoupled in primates

When body and brain weight are included, PC1 (capturing 96.4% of the variance) showed the evolution of increasing body weight together with the brain phenotypes in the 'diverse' group (*Figure 11a*).

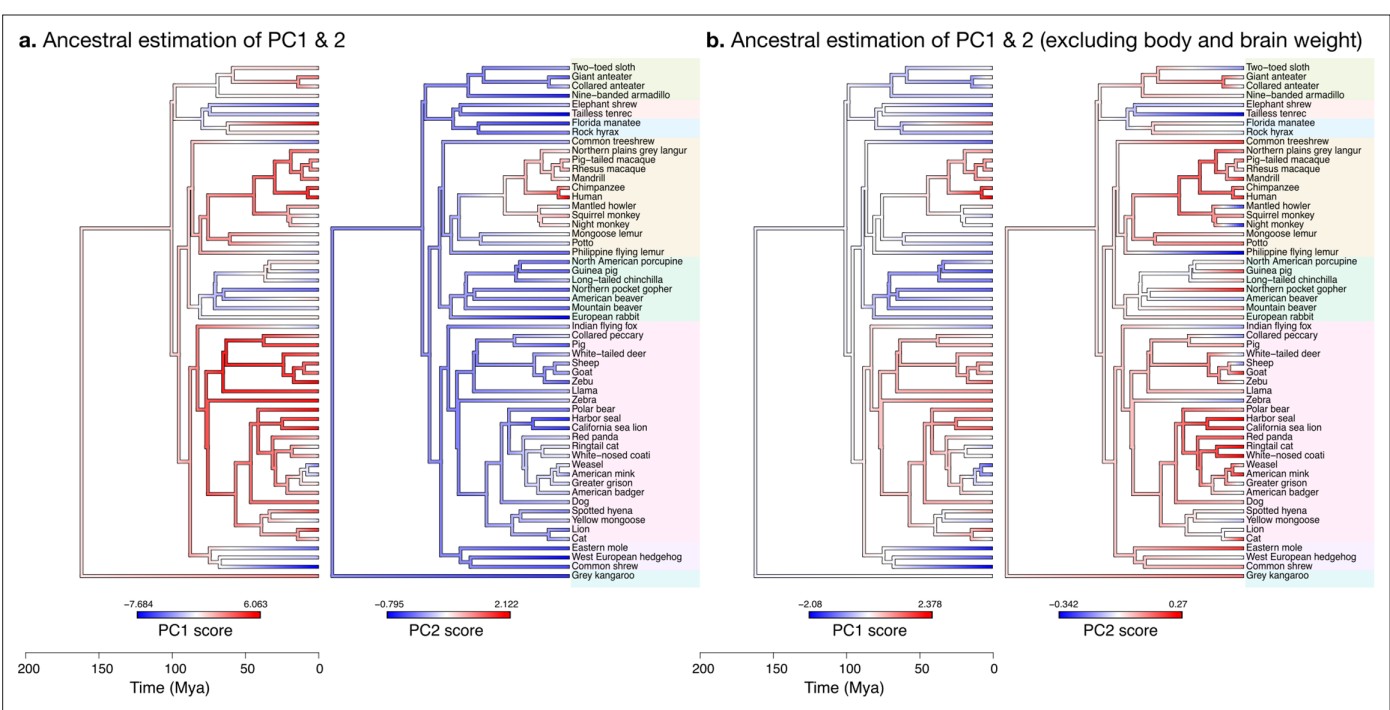

**Figure 11.** Estimation of ancestral neuroanatomical diversity patterns. Our analyses show a concerted change in body and brain size, and specific increase in cerebral and cerebellar volume in primates concomitant with an increased number of smaller folia. (**a**) Ancestral estimation of PC1 and PC2 for neuroanatomical variables plus brain volume. PC1 captures 96.4% of the variance, PC2 captures 2.4%. (**b**) Ancestral estimation of PC1 and PC2 for neuroanatomical variables, without brain volume. PC1 captures 91.4% of phenotypic variance, PC2 captures 3.6%.

These increases were estimated to appear first among the clade Scrotifera (harbour seal, zebu, pig, etc.) and later among primates. This set of phenotypes also decreased relative to the predicted common ancestor, for example, among Glires (rabbit, guinea pig, etc.) and Pilosa (eastern mole, common shrew, etc.). PC2 described the main deviations from this pattern, highlighting very particularly the group of primates, where brain sizes increased while body sizes stayed small. The common ancestor of mammals is predicted to have had a folded cerebrum with a correspondingly folded cerebellum, similar to that of a Rock hyrax.

### Diverse and stable phenotypes have decoupled in several branches of the phylogenetic tree

When excluding body and brain weight, PC1 loaded strongly on brain phenotypes from the 'diverse' group (*Figure 11b*). The ancestral estimation of PC1 (92.8% of the variance) appeared very similar to the previous one, highlighting the early concerted increase in cerebellar and cerebral size among Scrotifera, followed later by primates. The pattern of PC2, however, was now different from the previous PC2. It described an early decoupling of the group of 'diverse' and 'stable' brain phenotypes among primates, as well as a more recent decoupling among Scrotifera, especially Mustelida (harbour seal, ringtail cat, etc.). The cerebrum of the manatee is a well-known exception to the allometric relationship between brain size and folding: despite its large volume, it is almost completely lissencephalic. The evolutionary trajectory of the increase in manatee brain size (cerebrum and cerebellum) is clear in both PC1 ancestral estimations, however, the decoupling between diverse and stable phenotypes is not present.

## Discussion

*'Despite the impressive beauty of its wiring diagrams, the "neuronal machine" concept of the cerebellum remained vaguely defined as "a relatively simple machine devoted to some essential information processing." I was frustrated enough at the Salishan meeting to ask what else experimentalists would need to uncover before we would be able to understand the meaning of these wiring diagrams. Someone equally frustrated replied that the available diagrams were too simple to construct even a primitive radio, so more information was urgently needed before any meaningful model could be conceived.'- M Ito, The Cerebellum (2011)*

Whereas the circuits of the cerebral cortex are characterised by the presence of profuse re-entering loops (most connections of the cerebral cortex originate in other regions of the same cerebral cortex), the cerebellum exhibits an almost perfect feedforward structure, with an organisation whose regularity and simplicity has baffled generations of experimentalists and theoreticians. The impressive multi-scale regularity of cerebellar structure led to the idea of the 'neuronal machine' (*Eccles et al., 1967*). The quote from *Yamamoto et al., 2012*, illustrates the challenge of imagining how the obstinate repetition of a single interconnection pattern among a reduced number of components could lead to the complex cerebellar function. However, its impressive evolutionary conservation suggests that its function has to fulfil an important role, for motion and cognition (*Whiting and Barton, 2003*; *Ramnani, 2006*; *Barton and Venditti, 2014*; *Magielse et al., 2022*).

One possibility is that the complexity of cerebellar function should not be found at the level of the individual circuits – as would be the case for a radio – but that those circuits would provide the substrate over which complexity would emerge, as waves in the ocean or patterns in a vibrating Chladni plate. A deeper understanding of the mesoscopic and macroscopic scales of organisation would be probably more appropriate to understanding phenomena unique to this level of organisation, complementing the extensive exploration of microscopic circuits and molecular properties of cerebellar cells.

Our results provide a closer look into the nature of macroscopic cerebellar anatomy, its relationship with cerebral anatomy, its diversity across mammalian species, and its evolution. First, we showed that our computational neuroanatomy workflow, for which code and data have been made available open source, produces reliable results comparable with those of the literature. Second, we showed that the OU model, often used to model stabilising selection, is the one that fits phenotypic diversity the best. The OU model tended to dampen extreme variation, driving phenotypes towards the mean faster than the BM model would. Third, we showed that brain phenotypes segregated into a group of

'diverse' phenotypes, which increased over several orders of magnitude together with body size, and a second group of 'stable' phenotypes, which increased only slightly with brain size. Both groups were linked by significant negative partial correlations. The stable phenotypes were related to the local shape of cerebellar folding: folial width, folial perimeter, and thickness of the molecular layer. They changed over only ~0.5 order of magnitude between species with differences in brain size of ~2.5 orders of magnitude and difference in body size of ~11 orders of magnitude. Fourth, our allometric analyses confirmed the strong correlation between cerebellar and cerebral size (*Barton and Harvey, 2000*; *Barton, 2002*; *Whiting and Barton, 2003*; *Herculano-Houzel, 2010*; *Barton, 2012*; *Smaers et al., 2018*; *Ashwell, 2020*), and we extended these results to show that the same strong relationship holds for cerebellar folding: larger cerebella appear to be disproportionately more folded than smaller ones. Additionally, the allometric law for the increase of cerebellar folding was the same as for the increase of cerebral folding. Fifth, we estimated the evolutionary trajectories which may have led to the current phenotypic diversity. The main evolutionary trajectory showed the concerted increases and decreases in body size, brain size, cerebellar and cerebral size. Primates appeared as an outlier to this trend, with large brains relative to their body sizes. Focusing on brain phenotypes, our results showed a decoupling between diverse and stable phenotypes operating in several branches of the phylogenetic tree. This second pattern was, however, much more subtle than the global concerted increase in body and brain size.

Our results are influenced by a series of methodological choices. We will discuss the most important ones. First, we use a single cerebellar section and a single cerebral section for each species. More sections would increase the accuracy of our measurements, but would require a substantial manual segmentation effort. For our research question, having global measurements for many species was more relevant than having dense measurements for a few species. The previous work of *Sultan and Braitenberg, 1993*, is remarkable in this respect, providing beautiful representations of the unfolded cerebellar surface for 15 different species. However, their method produced smaller cerebellar surface areas than in the same species using a more precise method, suggesting that it underestimates the actual values (*Sereno et al., 2020*; *Zheng et al., 2023*), and hence the magnitude of cerebellar folding. Producing complete reconstructions of the cerebellar surface remains challenging, and as far as we know, only three cerebella have been precisely reconstructed: two from humans and one from a macaque (*Sereno et al., 2020*; *Zheng et al., 2023*). Second, we decided to use global measurements of brain anatomy. Although we computed folial width and folial perimeter for each folium (*Figure 3a–c*), and molecular layer thickness at numerous points for each brain section (*Figure 4*), we condensed all these local measurements into three median values. The cerebrum and the cerebellum show significant regional variability, but using local measurements would increase the number of phenotypes, largely exceeding the number of species. Fitting phylogenetic models with a significantly greater number of phenotypes than species is challenging (in particular, there is an ongoing debate regarding the use of OU models, see the work of *Cooper et al., 2016*; *Grabowski et al., 2023*). The use of penalisation techniques and constraints in the model's structure holds great promise and will enable the exploration of new hypotheses regarding the evolution of complex phenotypes (*Clavel et al., 2015*; *Clavel et al., 2019*). Finally, we decided to measure the raw material without aiming at correcting for shrinkage. Our reasoning was that if the relationships were sufficiently robust, we should still be able to see them despite shrinkage. This appeared to be the case: accounting for shrinkage should enhance the accuracy of our measurements making our correlations even stronger and our estimations more precise, but should not change our conclusions. Shrinkage correction is often performed using correction factors obtained from the literature, which introduces a new source of methodological variability. A more precise account of shrinkage would require, for example, acquiring MRI data before and after fixation for each species (see *Wehrl et al., 2015*). Sections could be then registered to the undeformed MRI, allowing us to estimate deformation due to slicing and shrinkage.

## The measurement of folding

Previous studies of cerebellar folding have relied either on a qualitative visual score (*Yopak et al., 2007*; *Lisney et al., 2008*) or a 'gyrification index' based on the method introduced by *Zilles et al., 1988*; *Zilles et al., 1989*, for the study of cerebral folding (*Iwaniuk et al., 2007*; *Cunha et al., 2020*; *Cunha et al., 2021*). Zilles's gyrification index is the ratio between the length of the outer contour of the cortex and the length of an idealised envelope meant to reflect the length of the cortex if it

were not folded. For instance, a completely lissencephalic cortex would have a gyrification index close to 1, while a human cerebral cortex typically has a gyrification index of ~2.5 (*Zilles et al., 1988*). This method has certain limitations, as highlighted by various researchers (*Germanaud et al., 2012*; *Germanaud et al., 2014*; *Stanković and Sejdić, 2019*; *Schaer et al., 2008*; *Toro et al., 2008*; *Heuer et al., 2019*). One important drawback is that the gyrification index produces the same value for contours with wide variations in folding frequency and amplitude, as illustrated in *Figure 3d*. In reality, folding frequency (inverse of folding wavelength) and folding amplitude represent two distinct dimensions of folding that cannot be adequately captured by a single number confusing both dimensions. To address this issue we introduced two measurements of folding: folial width and folial perimeter. These measurements can be directly linked to folding frequency and amplitude, and are comparable to the folding depth and folding wavelength we introduced previously for cerebral 3D meshes (*Heuer et al., 2019*). By using these measurements, we can differentiate folding patterns that could be confused when using a single value such as the gyrification index (*Figure 3d*). Additionally, these two dimensions of folding are important, because they can be related to the predictions made by biomechanical models of cortical folding, as we will discuss now.

## Modelling of cortical folding

Based on our results, we hypothesise that the process leading to cerebellar and cerebral folding is the same. In both cases, partial correlations showed strong direct links between section area and length. The allometric scaling slopes for section area and length – which describe the increase in degree of folding with size – were also the same. The conservation of folial width and perimeter is similar to the conservation of folding wavelength that we have reported previously in primates (*Heuer et al., 2019*).

The folding of the cerebral cortex has been the focus of intense research, both from the perspective of neurobiology (*Borrell, 2018*; *Llinares-Benadero and Borrell, 2019*; *Fernández and Borrell, 2023*) and physics (*Toro and Burnod, 2005*; *Tallinen et al., 2014*; *Kroenke and Bayly, 2018*). Current biomechanical models suggest that cortical folding should result from a buckling instability triggered by the growth of the cortical grey matter on top of the white matter core. In such systems, the growing layer should first expand without folding, increasing the stress in the core. But this configuration is unstable, and if growth continues, stress is released through cortical folding. The wavelength of folding depends on cortical thickness, and folding models such as the one by *Tallinen et al., 2014*, predict a neocortical folding wavelength which corresponds well with the one observed in real cortices. *Tallinen et al., 2014*, provided a prediction for the relationship between folding wavelength $\lambda$ and the mean thickness ($t$) of the cortical layer: $\lambda = 2\pi t(\mu/(3\mu_s))^{1/3}$. If we consider the stiffness of the cortex ($\mu$) and the substrate ($\mu_s$) to be similar ($\mu/\mu_s \approx 1$), we obtain in the case of humans, with an average cortical thickness of 2.5 mm (*Fischl and Dale, 2000*), a prediction of $\lambda$~10.9 mm. This corresponds well with the human folding wavelength of ~11.2 mm reported in *Heuer et al., 2019*.

In the case of the cerebellum, it has been suggested that the transitory external granular layer may play the role of the expanding cortical layer in the models of cerebral cortical folding (*Lawton et al., 2019*). During development, the external granular layer produces an astonishingly large number of small granule cells. These cells migrate towards the inside of the cerebellum past the Purkinje cells to constitute the granular layer (*Leto et al., 2016*). This idea was expanded by *Van Essen, 2020*; *Van Essen et al., 2018*, who proposed in particular that tangential tension along parallel fibres (axons of granule cells) constrains the orientation of folds and may be the reason for their accordion-like orientation.

We hypothesise alternatively that the expanding layer leading to cerebellar folding is the molecular layer, that growth is not driven by granule cell proliferation but by the growth of the dendritic arborization of Purkinje cells – the orientation of cerebellar folding resulting from their characteristic expansion: the dendritic trees of Purkinje cells grow mostly in 2D (*Kaneko et al., 2011*), which could result in anisotropic cortical growth leading to the observed parallel folds. Indeed, in the cerebrum the growth of dendritic trees seems to be the main factor leading to cortical expansion (*Welker, 1990*; *Rash et al., 2023*). By the end of neuronal migration from the ventricular and outer subventricular zones, the cerebral cortex is still largely lissencephalic (*Welker, 1990*). Most cerebral cortical folding starts after the end of neuronal migration and is concomitant with the development of cortico-cortical connectivity and the elaboration of neuronal dendritic trees. If the expansion of the molecular layer were the main driving force for cerebellar folding, we would expect folial width to be

proportional to the thickness of the molecular layer according to the same formula as for the cerebral cortex. This appears to be the case, and we observe that (1) in our data folial width and molecular layer thickness are, as expected, related by a significant positive partial correlation (*Figure 7c*); and (2) for a molecular layer thickness estimation across species of $t\sim200$ µm, the wavelength expected from Tallinen et al.'s formula of $\lambda\sim0.87$ mm corresponds well with our observed mean folial width across species of $\sim1$ mm.

## Role of cerebellar folding: a constraint for modularity?

A striking characteristic of cerebellar anatomy is the multi-scale nature of its folding: in small cerebella we can observe only a first level of folding, but a pattern of folding within folding appears progressively as cerebellar size increases. For example, in humans (*Figure 1*), we can distinguish at least three such levels of folding. The addition of new levels of folding could be the reason why we observed relatively smaller folial width in larger cerebella (*Figure 7c*) such as those of humans, as new smaller folds develop atop existing ones. In the human cerebrum, folding has been reported to show one level of folds on top of folds, or 'frequency doubling' (*Germanaud et al., 2012*). This phenomenon has also been observed in swelling gels and mechanical models of folding (*Mora and Boudaoud, 2006*). We expect that such mechanical models should be able to produce additional levels of folding if more growth were allowed or if cortices were made thinner.

For both the cerebellum and the cerebrum, the thickness of the expanding layer appears to be very stable compared with the large diversity in total volume, which could reflect to some extent a conservation of local circuits across species. Regarding the cerebellum, we could speculate that the hierarchical organisation induced by folding promotes a similar hierarchical functional organisation on top of conserved local circuits. The addition of an increasing number of folial 'modules' and the constitution of a nested hierarchy of super modules should induce a similar modularity of white matter connections. Cerebellar folding could then constrain the cerebellum to develop a regular, hierarchical pattern of variation in fibre length with an associated modular pattern in timing of neuronal spikes. Then, through synaptic plasticity, this could lead to a preferential potentiation of neurons within the same folium, and next within their super-folium, etc. The timing of the development of cerebellar folding could also have an influence on the constitution of cerebellar networks, reinforcing the influence of early modules (trees) on late modules (leafs).

## Acknowledgements

We thank Carol MacLeod, Fahad Sultan, David DiGregorio, and Maria Castelló for discussions on cerebellar evolution and cerebellar folding, and Liam Revell for help and advice with phylogenetic analyses. Funded by project NeuroWebLab (ANR-19-DATA-0025), DMOBE (ANR-21-CE45-0016), the European Union's Horizon 2020 research and innovation programme under the Marie Skłodowska-Curie grant agreement No 101033485 (KH Individual Fellowship), and the STIC-AmSud programme project STIC-AMSUD+CLANN 22-STIC-03.

## Additional information

### Funding

| Funder | Grant reference number | Author |
|---|---|---|
| European Commission | 101033485 | Katja Heuer |
| Agence Nationale de la Recherche | ANR-19-DATA-0025 | Katja Heuer |
| Agence Nationale de la Recherche | ANR-21-CE45-0016 | Katja Heuer |

The funders had no role in study design, data collection and interpretation, or the decision to submit the work for publication.

## Author contributions
Katja Heuer, Roberto Toro, Conceptualization, Resources, Data curation, Software, Formal analysis, Supervision, Funding acquisition, Validation, Investigation, Visualization, Methodology, Writing – original draft, Project administration, Writing – review and editing; Nicolas Traut, Software, Methodology; Alexandra Allison de Sousa, Writing – review and editing; Sofie Louise Valk, Data curation, Writing – review and editing; Julien Clavel, Software, Visualization, Methodology, Writing – review and editing

## Author ORCIDs
Katja Heuer https://orcid.org/0000-0002-7237-0196
Alexandra Allison de Sousa http://orcid.org/0000-0003-2379-3894
Sofie Louise Valk https://orcid.org/0000-0003-2998-6849
Roberto Toro http://orcid.org/0000-0002-6671-858X

## Ethics
Human subjects: The human dataset used has been made openly available through the BigBrain project.

## Decision letter and Author response
Decision letter https://doi.org/10.7554/eLife.85907.sa1
Author response https://doi.org/10.7554/eLife.85907.sa2

---

# Additional files

## Supplementary files
• Supplementary file 1. List of species included in this study, sorted alphabetically by orders and binomial name. Excluded$_{W\&P}$: excluded from width and period analyses. Excluded$_{Th}$: excluded from analyses of molecular layer thickness.

• Supplementary file 2. Partial correlations example together with further details on our power analysis (also available as a Jupyter notebook with the accompanying source code).

• Supplementary file 3. Author response image 2.

• MDAR checklist

## Data availability
All data analysed during this study is openly available at https://microdraw.pasteur.fr/project/brainmuseum-cb. Code for reproducing our analyses and our figures is openly available at https://github.com/neuroanatomy/comp-cb-folding (copy archived at *Heuer et al., 2023*).

The following dataset was generated:

| Author(s) | Year | Dataset title | Dataset URL | Database and Identifier |
|---|---|---|---|---|
| Heuer K, Traut N, de Sousa AA, Valk S, Clavel J, Toro R | 2023 | Cerebellar and cerebral contours | https://microdraw.pasteur.fr/project/brainmuseum-cb | MicroDraw, /project/brainmuseum-cb |

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
