## [Editor Report]

This fundamental study gives novel insight into the folding diversity of the cerebellum compared to the cerebrum among 56 mammalian species. Based on impressive data, a variety of convincing analyses are performed, in particular for anatomical descriptions, phylogenetic comparisons and allometry investigations. This study will be of great interest to biologists, especially evolutionary and comparative neuroscientists, and physicists interested in biomechanics, as these observations provide a basis for models of brain folding mechanisms.

---

## [Decision Letter]

**Decision letter after peer review:**

Thank you for submitting your article "Diversity and evolution of cerebellar folding in mammals" for consideration by *eLife*. Your article has been reviewed by 3 peer reviewers, including Jessica Dubois as the Reviewing Editor, and the evaluation has been overseen by Floris de Lange as the Senior Editor. The following individuals involved in the review of your submission have agreed to reveal their identity: Felipe Cunha (Reviewer # 1); Héloïse de Vareilles (Reviewer #2); Robert A Barton (Reviewer #3).

Essential revisions:

All reviewers acknowledged the importance and originality of your study. But they also raised a number of important concerns to consider.

The most critical one concerns the statistical analyses performed (see Reviewer #3's comments). Further evidence should be provided regarding the superiority of the Ornstein-Uhlenbeck model. Besides, the objectives and relevance of the partial correlations and principal component analyses to all included variables should also be better argued given the possible over-fitting and the underlying theoretical hypotheses which are not always clear.

The reviewers were also concerned about the global presentation of the manuscript. Additional description of some methods is required. The different results should be better linked, and potentially reordered. The discussion should also be reviewed in depth. Better highlighting the methodological novelties should also be considered.

We hope that you will be able to take all these general comments into account, as well as the more detailed comments of the reviewers (see below).

*Reviewer #2 (Recommendations for the authors):*

This study is both rich and generous, and I thank the authors for the beautiful work presented and shared with the community. The investigation of the cerebellum is both essential to our general understanding of the brain and under-explored, making the efforts developed in this article essential.

I find the analyses well-led. Nevertheless, I think the reach of the current work would gain from a better emphasis on why the analyses have been conducted and what they imply. There are also aspects of the structure of the article which I find unconventional and potentially confusing for the reader.

I will address my general recommendations following the current structure of the article.

I will not address more specific concerns (specific wording, punctuation marks…) that I have in this step of the review for two reasons:

1) I believe the more general recommendations may lead to a restructuring of the article which would make some of the specific concerns irrelevant.

2) The current draft does not include line numbers, making it challenging to address line-specific recommendations (please add line numbering to the next version if possible).

Introduction:

– I have identified that the last three paragraphs match an Aims (bottom of p.2) – Methods (first paragraph p.3) – Results (second paragraph p.3). Then, the buckling models consideration of the Methods paragraph is rather an aim than a method and should be addressed in the previous paragraph.

– In my opinion, the results should not be addressed in the introduction, as there is yet no information to support them. They should be summarized at the start of the discussion instead.

Methods:

– Histological data: the microdraw.pasteur.fr/project/brainmuseum-cb webpage was not publicly available at the time of writing of this review (Not authorized to view project – 14.03.23). Please ensure that you make it available.

– Histological data: what are the anatomical criteria to identify the mid-cerebellar slice? Do these criteria ensure that you are looking at comparable structures between species? (this can either be addressed in methods or in discussion).

– Histological data: exclusion threshold – It would be nice if you included it in supp. Mat. An example of images that are just under and just over this threshold for the reader to get a grasp of the image quality requirement of the study.

– Measurement of section area and length of the cerebral and cerebellar cortex: The implications of measurements not being modified to account for shrinkage should be briefly addressed in the discussion.

– Figure 2/Table 1: the cerebellar section area/length would gain from its boundary being more explicitly defined in Figure 2's legend: is it the black contour in both cases? Is there a difference between "cerebellar mid-slice section" and "cerebellar pial surface of mid-slice section" as a boundary for those two items?

– Measurement of molecular layer thickness: From "We applied a Laplacian smoothing […]" to "from the outside of the image." Is it the same concept with two different wordings? Maybe blend the two sentences together or delete one as it may appear that two different gradients are computed.

– Figure 4: I find image 4c confusing: is it related to images 4a and 4b? If so, why are the red dots creating such an angular curve when the black region's boundary appears very smooth on 4a and 4b? Please add more details to the legend.

– Please address all the numerical analyses led in the Results section within the methods section.

1) It would be helpful if some of the more complex types of results were introduced in the methods section, such as partial correlations conditional to all remaining measurements, or multivariate allometric scaling slopes.

2) It would be helpful if the impact of genetic model choice on the other analyses was clarified (as in, which subsequent analyses rely on this model selection).

Results:

I could not find the common thread linking all the results together. Can you either make the logical structure of the results explicit or change it so that it follows a logical order? As an example, it could go from simpler to more complex analyses.

In addition, I would move some results to the discussion. This could lead to the following structure: – statement of numerical results (including section areas if relevant and mean thickness of the molecular layer) – correlation structure – allometries – genetic models assessment – phylogenetic PCA – Ancestor character state estimation.

I suggest moving comparisons with anterior studies in the discussion, both for the very first and very last section of the results.

– Correlation structure: p.9 – please detail partial correlations and their meaning in methods.

– Correlation structure: p.9 – please detail "phylogenetic" PCA in methods.

– Correlation structure: p.10 – The PCA analyses should have their own subsection.

– Correlation structure: p.11 – please detail allometric slopes and multivariate allometric slopes in methods (are they used anywhere?).

– Figure 8: the method for creating coloured clusters for each group should be briefly explained, maybe in the methods section?

– Allometry: p.14 – you could re-quote the grey area in figure 10 when mentioning significance since it is not very visible and it may not be obvious to the reader that it is present in the figure.

Discussion:

In my opinion, the discussion needs to be thoroughly reworked. The current content of the discussion is very general and seems to address questions related to the work proposed rather than the work in itself. Discussing related topics is interesting and relevant, but should only take a subsection of the discussion.

Here is a list of subsections that I would expect in a discussion (the order can be changed):

– summary of the results demonstrated.

– considerations about the work in a link to the state of the art (maybe 1st section of the results, considerations about coherence with the Tallinen model, and considerations about limits of the Sultan and Braitenberg pioneer study).

– discussion of methodological decisions and potential limitations (not taking into account shrinkage, comparability of mid-cerebellar slice between species, the necessity of a sophisticated method for measurement of molecular layer thickness, …).

– discussion about the interpretation of the results, ideally creating an overall link about the different results and what they tell. This is essential to convey the overall meaning of the results reported.

– relation of the current report to broader questions and considerations (has to come after the interpretation of results, or it can be confusing for the reader).

*Reviewer #3 (Recommendations for the authors):*

Results/stats

– Please either remove the OU models or at least discuss these issues and justify conclusions about model fit with more caution.

– Please provide more justification for variable inclusion in the partial correlation and PC analyses, clarify the theoretical motivation for and inferences from the analyses, and discuss statistical issues of statistical power and collinearity.

– Please consider testing the hypothesis of cortico-cerebellar correlated evolution by running models something like [cerebellar measure ~ cortical measure + rest of the brain measure]; and/or if this is not possible, clarify the claims you are making about correlated evolution of cortex and cerebellum (why the patterns indicate something more specific than overall size).

– Figure 6 a: please at least present the CIs on the coefficients to show that the observed values include the predicted one.

Referencing/discussion/attribution of previous findings:

– Please revise/discuss/add citations as appropriate, based on my comments and references provided in the public review

[Editors’ note: further revisions were suggested prior to acceptance, as described below.]

Thank you for resubmitting your work entitled "Diversity and evolution of cerebellar folding in mammals" for further consideration by *eLife*. Your revised article has been evaluated by Floris de Lange (Senior Editor) and a Reviewing Editor.

The manuscript has been improved but there are some remaining issues that need to be addressed, as outlined below.

Among the reviewers' recommendations (detailed below), the main criticisms are as follows:

– Following the second comment from the third reviewer (see below), all reviewers and the editor agree that the partial correlation analyses may be problematic in view of the sample size, and that they need to be better justified and interpreted to guarantee the robustness of the results.

– The justification of the phylogenetic model still raises questions.

– The section about the measurement of molecular layer thickness could be further clarified.

*Reviewer #2 (Recommendations for the authors):*

I thank the authors for addressing most of the concerns issued thoroughly. Overall, the aims, methods and meanings of the results are clearer and my understanding was hence improved. In my opinion, only two points remain unresolved.

Unresolved points:

I. Accessibility of the Brain Museum Cerebellum project:

When trying to access it, I still get an error: "Not authorized to view project".

II. I still don't understand the section about the measurement of molecular layer thickness. (p.6)

This section has not been modified since the previous version of the article, and I find it challenging because it presents a rather complex methodology with not very clear context. In particular, the sentence "In mature cerebella, the molecular layer appears as a light band followed by the granular layer which appears as a dark band" with no additional comment is confusing in my opinion: are you dealing with only mature cerebella which makes the boundary between molecular and granular layer easy to detect or is it this not-so-obvious boundary that makes this complex method necessary? what does the boundary detection in Figure 4.c represent regarding Figure 4.b? I guess the molecular layer thickness results from averaging the length of the profile lines from figure 4.b?

To sum up, I have difficulties understanding what the problem is and hence what the method solves. I think this paragraph would gain from being more explicit in the reasons for both the whole and subparts of the process.

Apart from that, I found the current version much clearer than the previous one.

*Reviewer #3 (Recommendations for the authors):*

In general, I am happy with the responses of the authors and the changes they have made. There are still two statistical issues requiring a response:

1. The justification for the OU model. Grabowski et al. do not actually demonstrate that the critique by Cooper et al. is misleading – mainly they just show that Cooper et al. were correct! That is not to say that the OU model can never be justified (although some query its basis in biological reality), but that instances of its superiority over BM need to be treated very cautiously. I don't think their simulation solves the problem – the reason the OU is biased is because it does not conform to the data distributions assumptions of AIC (which has strong assumptions about normality or AICc) – this would be the same for other tree transforms like δ (Pagel). This results in the OU being statistically favoured when the 'pull' parameter is so weak it is essentially the same as the Brownian model.

2. The authors reject the criticism of overfitting by stating that their analyses do not involve fitting models. This is technically correct in strict terms but misleading. Exactly the same issues around small sample sizes per parameter (or variable) arise in partial correlation networks as in fitting a multiple regression model. Partial regression coefficients and partial correlation coefficients are susceptible to the same problem- neither can be estimated accurately without sufficient statistical power. Say you have 20 variables and 20 data points – no one would (hopefully) run a partial correlation on the 20 variables, or even 10 variables, or 5 – maybe just 2 could be justified. Here, the authors have 9 variables and 56 data points – with only 6 data points per variable, they don't have the statistical power to tease apart the relationships as they imply. Added to this, even with large samples, throwing in a bunch of variables without some explicit model that you are testing does indeed result in a correlational salad. Typically, such a partial correlation network would be liable to instability, and the more variables you add the harder it is to make sense of the individual relationships between them. Their explanation of how a negative correlation can arise between variables after controlling for their relationship to other variables is not necessary – this is obvious. What is not so obvious is why in principle there should be a negative (partial) correlation within their network – what is its interpretation? If they started with a model that predicted this or had some compelling explanation, I would be more impressed, but as it is I suspect it is an artefact for the reasons stated above.

---

## [Author Response]

Essential revisions:All reviewers acknowledged the importance and originality of your study. But they also raised a number of important concerns to consider.The most critical one concerns the statistical analyses performed (see Reviewer #3's comments). Further evidence should be provided regarding the superiority of the Ornstein-Uhlenbeck model.

As suggested by Reviewer #3, we evaluated our ability to decide correctly for the Brownian Motion model over the Ornstein-Uhlenbeck model by simulating multivariate correlated phenotypes (9 variables) for 56 species using our phylogenetic tree. Over 1000 simulations, the Brownian Motion model was correctly selected 99.3% of the times (see the answer to Reviewer #3 for more details). This additional analysis is added to the revised version of the manuscript.

Besides, the objectives and relevance of the partial correlations and principal component analyses to all included variables should also be better argued given the possible over-fitting

We have now added a more detailed description of the objectives of our analysis of partial correlations, principal component analysis, and allometry. We have aimed at providing an intuitive explanation. Neither partial correlations nor principal components rely on the fitting of a model, so there is no possible over-fitting. We hear, however, Reviewer #3’s concern about the number of phenotypes per species becoming too large for the complexity of the models, and the temptation to run arbitrary comparisons. We share that concern, and the primary objective of these two methods is to avoid the “correlation salad” evoked by the Reviewer by looking at global multivariate patterns of covariation. Our simulations also provide evidence that the complexity of our model remains adequate.

and the underlying theoretical hypotheses which are not always clear.

We have completely re-structured our manuscript to make our theoretical standpoint more clear, providing a lengthier discussion of our methodological approach, and a more clear structure of our results and discussion, as suggested by the Reviewers. Our approach is not to test particular hypotheses, but to provide a characterisation of the phenomenon of cerebellar brain folding and its relationship with the folding of the cerebrum, as we indicate in the introduction.

From this characterisation we were able to advance hypotheses (presented in the discussion) which could be tested in the future.

The reviewers were also concerned about the global presentation of the manuscript. Additional description of some methods is required.

The revised manuscript has an expanded Methods section including introductions to our methods as well as illustrations aiming at highlighting their meaning and relevance. We introduce, as suggested by the Reviewers, an introduction to the measurement of folding and the advantages of our method (illustrated in Figure 3d), an introduction to the phylogenetic comparative models that we used, including an illustration showing their importance for the study of comparative data (Figure 5b), an introduction to the analysis of partial correlations with an example that aims at making their meaning more intuitive, especially concerning negative partial correlations, an introduction to principal components analysis for phylogenetic data, and an introduction to the study of allometry for bivariate and multivariate data, with examples from the literature. We have aimed at finding a good compromise between making a clear exposition, while keeping the size of the methods section reasonable. The examples that we have introduced can be reproduced and explored further in Jupyter notebooks which are included in the accompanying Github repository.

The different results should be better linked, and potentially reordered.

We have re-structured the Results section and provided a more clear description of their links, following the advice of Reviewer #2. Our Results section starts now with a guiding paragraph, which explains the links between the results in their logical sequence.

The discussion should also be reviewed in depth.

This is now also the case. We have integrated the advice of all Reviewers, adapting our structure to the one they proposed, in particular, including a more detailed description of the implications and limitations of our methodological decisions, and a summary discussion of each of our results.

Better highlighting the methodological novelties should also be considered.

The methods section has been expanded to include more detail on our methodology, which serves to highlight the novelty and relevance of our approach.

We hope that you will be able to take all these general comments into account, as well as the more detailed comments of the reviewers (see below).Reviewer #2 (Recommendations for the authors):This study is both rich and generous, and I thank the authors for the beautiful work presented and shared with the community. The investigation of the cerebellum is both essential to our general understanding of the brain and under-explored, making the efforts developed in this article essential.I find the analyses well-led. Nevertheless, I think the reach of the current work would gain from a better emphasis on why the analyses have been conducted and what they imply. There are also aspects of the structure of the article which I find unconventional and potentially confusing for the reader.I will address my general recommendations following the current structure of the article.I will not address more specific concerns (specific wording, punctuation marks…) that I have in this step of the review for two reasons:1) I believe the more general recommendations may lead to a restructuring of the article which would make some of the specific concerns irrelevant.2) The current draft does not include line numbers, making it challenging to address line-specific recommendations (please add line numbering to the next version if possible).

Unfortunately, there doesn’t seem to be a way of adding line numbers in Google docs, and moving to LibreOffice completely broke the layout… However, the information the Reviewer provided was very clear and we did not have difficulties finding the sections in the paper that were being discussed.

Introduction:– I have identified that the last three paragraphs match an Aims (bottom of p.2) – Methods (first paragraph p.3) – Results (second paragraph p.3). Then, the buckling models consideration of the Methods paragraph is rather an aim than a method and should be addressed in the previous paragraph.

The revised version of the manuscript provides additional context which highlights the novelty of our approach, in particular concerning the measurement of folding and the use of phylogenetic comparative models. The limitations of the previous approaches are stated more clearly, and illustrated in Figures 3 and 5.

Following the Reviewer’s advice, we have thoroughly restructured our results and Discussion section.

– In my opinion, the results should not be addressed in the introduction, as there is yet no information to support them. They should be summarized at the start of the discussion instead.

The brief mention of buckling models at the end of the introduction provides our justification for our measurement of folial width and molecular layer thickness. The idea of that short summary of our results is to allow readers to decide if they want to keep reading further. They are not described in any detail, just listed. A more detailed summary is provided in the discussion, as suggested by the Reviewer.

Methods:– Histological data: the microdraw.pasteur.fr/project/brainmuseum-cb webpage was not publicly available at the time of writing of this review (Not authorized to view project – 14.03.23). Please ensure that you make it available.

The MicroDraw project is now public.

– Histological data: what are the anatomical criteria to identify the mid-cerebellar slice? Do these criteria ensure that you are looking at comparable structures between species? (this can either be addressed in methods or in discussion).

We realised that the term mid-slice may have been confusing, and we now call it mid-section throughout the manuscript. It is simply the section in the middle of the cerebellum: there is one section that is the first one to contain the cerebellum, another one that is the last one to contain the cerebellum, this is the one that is in the middle (or the nicest of the 2 middle ones if there’s an even number of slices). The exact index of the sections is included in the repository with our data and code.

– Histological data: exclusion threshold – It would be nice if you included it in supp. Mat. An example of images that are just under and just over this threshold for the reader to get a grasp of the image quality requirement of the study.

Our whole project is now public, giving access to the raw images, and allowing for easy visual inspection of their differences.

– Measurement of section area and length of the cerebral and cerebellar cortex: The implications of measurements not being modified to account for shrinkage should be briefly addressed in the discussion.

As suggested by the Reviewer, the revised version of our manuscript includes a more detailed description of our methodological decisions and their limitations. We discuss in particular the issue of shrinkage:

“Finally, we decided to measure the raw material without aiming at correcting for shrinkage. Our reasoning was that if the relationships were sufficiently robust, we should still be able to see them despite shrinkage. This appeared to be the case: accounting for shrinkage should enhance the accuracy of our measurements making our correlations even stronger and our estimations more precise, but should not change our conclusions. Shrinkage correction is often performed using correction factors obtained from the literature, which introduces a new source of methodological variability. A more precise account of shrinkage would require, for example, acquiring MRI data before and after fixation for each species (see Wehrl et al. 2015). Sections could be then registered to the undeformed MRI, allowing us to estimate deformation due to slicing and shrinkage.”

– Figure 2/Table 1: the cerebellar section area/length would gain from its boundary being more explicitly defined in Figure 2's legend: is it the black contour in both cases? Is there a difference between "cerebellar mid-slice section" and "cerebellar pial surface of mid-slice section" as a boundary for those two items?

To avoid terminological confusion we now use mid-section throughout the manuscript. The contour we drew is the one that appears in black (we now indicate this in Figure 2’s legend), with the inside of the contour being filled in translucent red.

– Measurement of molecular layer thickness: From "We applied a Laplacian smoothing […]" to "from the outside of the image." Is it the same concept with two different wordings? Maybe blend the two sentences together or delete one as it may appear that two different gradients are computed.

We refer to 2 different things. Laplacian smoothing produces a potential field which has a gradual change. The gradient (the mathematical operation) gives the direction of maximum change along this potential field. We changed the formulation from “We applied a Laplacian smoothing to the masked image, which created a gradient going from (…)” to “We applied a Laplacian smoothing to the masked image, which created a gradual change in grey level going from…”

– Figure 4: I find image 4c confusing: is it related to images 4a and 4b? If so, why are the red dots creating such an angular curve when the black region's boundary appears very smooth on 4a and 4b? Please add more details to the legend.

The Figure 4a-b show a small excerpt of the whole contour, which may give the impression of regularity. Figure 4c, shows all the profiles of the section. The revised version of the manuscript includes a more detailed legend.

– Please address all the numerical analyses led in the Results section within the methods section.1) It would be helpful if some of the more complex types of results were introduced in the methods section, such as partial correlations conditional to all remaining measurements, or multivariate allometric scaling slopes.2) It would be helpful if the impact of genetic model choice on the other analyses was clarified (as in, which subsequent analyses rely on this model selection).

We now provide an introduction to partial correlations, with an example illustrating their meaning, especially for the negative ones. We also provide an introduction to allometry, phylogenetic models, and phylogenetic PCA. All this serves to clarify the impact and the necessity of phylogenetic models.

Results:I could not find the common thread linking all the results together. Can you either make the logical structure of the results explicit or change it so that it follows a logical order? As an example, it could go from simpler to more complex analyses.

The Results section in the revised manuscript includes a paragraph providing a rationale for the presentation of the results, as suggested by the Reviewer.

In addition, I would move some results to the discussion. This could lead to the following structure: – statement of numerical results (including section areas if relevant and mean thickness of the molecular layer) – correlation structure – allometries – genetic models assessment – phylogenetic PCA – Ancestor character state estimation.

As suggested by the Reviewer, we have displaced the allometry subsection after the subsection on correlation structure. The new structure is as follows: (1) validation of measurements, (2) evolutionary model selection, (3) correlation structure, (4) allometry, (5) ancestral phenotype estimation. The rationale for this ordering is provided in the first paragraph of the Results section.

I suggest moving comparisons with anterior studies in the discussion, both for the very first and very last section of the results.

We followed the advice of the Reviewer and moved the consideration on mechanical models to the discussion. However, we believe that the “validation” section has to come at the beginning, because it aims at showing that our methodology of measurement is appropriate.

– Correlation structure: p.9 – please detail partial correlations and their meaning in methods.

An introduction to partial correlations is now included in the methods.

– Correlation structure: p.9 – please detail "phylogenetic" PCA in methods.

An introduction to phylogenetic models and in particular their use in phylogenetic principal component analysis is now included in the methods.

– Correlation structure: p.10 – The PCA analyses should have their own subsection.

PCA is a method for the study of correlation structure, and as such the correlation structure section is its natural place. We have aimed at making the link between PCA, partial correlations and correlations more clear in the revised text.

– Correlation structure: p.11 – please detail allometric slopes and multivariate allometric slopes in methods (are they used anywhere?).

We now include a more detailed introduction of allometry, bivariate and multivariate. We indicate that multivariate slopes are used for all our allometric analyses, and we included a new table, Table 3, with the loadings of PC1 which describe the multivariate allometric relationships.

– Figure 8: the method for creating coloured clusters for each group should be briefly explained, maybe in the methods section?

The coloured clusters are only used to facilitate the visualisation of the groups. They are computed by the ggscatter package in R by fitting a normal distribution to the data points, and selecting a threshold. The colour scheme is indicated in each figure, and corresponds to the one used in Figure 5 and Figure 11.

– Allometry: p.14 – you could re-quote the grey area in figure 10 when mentioning significance since it is not very visible and it may not be obvious to the reader that it is present in the figure.

We have done as suggested by the Reviewer. We have also explicitly added the numeric CI ranges to all figures, in addition to their graphical representation.

Discussion:In my opinion, the discussion needs to be thoroughly reworked. The current content of the discussion is very general and seems to address questions related to the work proposed rather than the work in itself. Discussing related topics is interesting and relevant, but should only take a subsection of the discussion.

The discussion has been completely revised and reorganised, addressing our questions, results and hypotheses more precisely.

Here is a list of subsections that I would expect in a discussion (the order can be changed):– summary of the results demonstrated.– considerations about the work in a link to the state of the art (maybe 1st section of the results, considerations about coherence with the Tallinen model, and considerations about limits of the Sultan and Braitenberg pioneer study).– discussion of methodological decisions and potential limitations (not taking into account shrinkage, comparability of mid-cerebellar slice between species, the necessity of a sophisticated method for measurement of molecular layer thickness, …).– discussion about the interpretation of the results, ideally creating an overall link about the different results and what they tell. This is essential to convey the overall meaning of the results reported.– relation of the current report to broader questions and considerations (has to come after the interpretation of results, or it can be confusing for the reader).

Following the advice of the Reviewer, the discussion is now organised as follows: (1) general context for our question, (2) detailed summary of our results, (3) discussion on methodological decisions and limitations, (4) discussion on the measurement of folding, highlighting the relevance of our new approach, (5) discussion on models of cortical folding, stating our hypotheses and supporting them through our results, (6) a consideration of our results to the broader question of the potential role of cerebellar folding.

Reviewer #3 (Recommendations for the authors):Results/stats– Please either remove the OU models or at least discuss these issues and justify conclusions about model fit with more caution.

We have supported our result on the selection of the OU model with additional simulations.

– Please provide more justification for variable inclusion in the partial correlation and PC analyses, clarify the theoretical motivation for and inferences from the analyses, and discuss statistical issues of statistical power and collinearity.

We now provide a more detailed justification for our analyses including and excluding body weight and brain weight. Our introduction to partial correlations and PCA in the methods section should help clarify their utility for addressing the problem of collinearity.

– Please consider testing the hypothesis of cortico-cerebellar correlated evolution by running models something like [cerebellar measure ~ cortical measure + rest of the brain measure]; and/or if this is not possible, clarify the claims you are making about correlated evolution of cortex and cerebellum (why the patterns indicate something more specific than overall size).

Our introduction to partial correlation analysis should now clarify the similarity with the suggestion of the Reviewer (i.e., considering a bivariate relationship while covarying the rest).

– Figure 6 a: please at least present the CIs on the coefficients to show that the observed values include the predicted one.

95% confidence intervals have now been added for the slopes in the linear regressions and orthogonal regressions. We have also added p-values for all plots, as well as for the partial correlations in the diagram in Figure 7c.

Referencing/discussion/attribution of previous findings:– Please revise/discuss/add citations as appropriate, based on my comments and references provided in the public review

We have added the references suggested by the Reviewer.

[Editors’ note: what follows is the authors’ response to the second round of review.]

The manuscript has been improved but there are some remaining issues that need to be addressed, as outlined below.Among the reviewers' recommendations (detailed below), the main criticisms are as follows:– Following the second comment from the third reviewer (see below), all reviewers and the editor agree that the partial correlation analyses may be problematic in view of the sample size, and that they need to be better justified and interpreted to guarantee the robustness of the results.

We have conducted additional analyses which demonstrate that our results using partial correlation analyses are robust. Our simulations show that we have excellent statistical power to detect even medium size partial correlations, due to the strength of the allometric pattern that we describe. This is referred to in the manuscript and its implementation is detailed in the Supplementary Methods.

– The justification of the phylogenetic model still raises questions.

We provide additional arguments to support our approach, and references to relevant publications. Despite the OU model being widely used and implemented by all major packages for phylogenetic comparative analyses, we recognise that it is important to expose the criticism that this approach has received, in particular in the literature mentioned by Reviewer #3. This is especially important given that many potential readers of our work may not be familiar with phylogenetic comparative methods. We have highlighted this in our revised manuscript.

– The section about the measurement of molecular layer thickness could be further clarified.

We have integrated the suggestions of Reviewer #2, and clarified the outstanding points.

Reviewer #2 (Recommendations for the authors):I thank the authors for addressing most of the concerns issued thoroughly. Overall, the aims, methods and meanings of the results are clearer and my understanding was hence improved. In my opinion, only two points remain unresolved.Unresolved points:I. Accessibility of the Brain Museum Cerebellum project:When trying to access it, I still get an error: "Not authorized to view project".

The project is publicly available now.

II. I still don't understand the section about the measurement of molecular layer thickness. (p.6)This section has not been modified since the previous version of the article, and I find it challenging because it presents a rather complex methodology with not very clear context. In particular, the sentence "In mature cerebella, the molecular layer appears as a light band followed by the granular layer which appears as a dark band" with no additional comment is confusing in my opinion: are you dealing with only mature cerebella which makes the boundary between molecular and granular layer easy to detect or is it this not-so-obvious boundary that makes this complex method necessary?

We may have been providing irrelevant information, which may be the cause of the confusion. “Mature cerebella” makes reference to cerebellar development – which is not the focus of our manuscript. During cerebellar development, there is a transient layer of cell nuclei close to the surface, composed of migrating granular cells. Hence, during cerebellar development, there is not only one but two layers of neuronal nuclei (dark), one external and one internal. Migrating granule cells move from the external to the internal layer, which is why in mature cerebellar cortex only one layer of neurones is present (i.e., the internal dark band observed in histological data). The aim of our algorithm is then to detect the border of this dark layer. We have removed the mention to “mature cerebella” to prevent confusion.

The molecular layer of the cerebellum appears as a light band followed by the granular layer which appears as a dark band.

what does the boundary detection in Figure 4.c represent regarding Figure 4.b? I guess the molecular layer thickness results from averaging the length of the profile lines from figure 4.b?To sum up, I have difficulties understanding what the problem is and hence what the method solves. I think this paragraph would gain from being more explicit in the reasons for both the whole and subparts of the process.Apart from that, I found the current version much clearer than the previous one.

Figure 4a shows our manually drawn surface. Figure 4b shows profile lines going from the manually drawn surface in Figure 4a to the granular layer. Each of the columns in Figure 4c corresponds to one of the profile lines in Figure 4b. Figure 4b shows just an excerpt of the whole cerebellar slice, and then just a subset of all profile lines. Figure 4c, on the contrary, shows all the profile lines for that cerebellar slice. The vertical coordinate of each profile line in Figure 4c represents the distance from the cerebellar surface; the horizontal coordinate represents the gradient (in mathematical terms) of the profile grey level. When a value changes little, the gradient is small, but when there is a large change, the gradient is large. We use this to detect the border between the molecular layer (light) and the granular layer (dark), which produces a large change in gradient. In Figure 4c these points are indicated with a red spot. We have added additional precision to our explanations.

“We estimated the thickness of the molecular layer automatically (Figure 4). We processed the histological images to convert them to grey levels, denoise them and equalise their histograms, using functions from the Scikit-image package (van der Walt et al. 2014). Starting from the manual segmentation (blue line in Figure 4a), we created a binary mask and masked out the non-cerebellar regions (for example, the neighbouring cerebrum). The molecular layer of the cerebellum appears as a light band followed by the granular layer which appears as a dark band. We applied a Laplacian smoothing to the masked image, which created a gradual change in grey level going from white close to the surface of the cerebellum to black toward the granular layer. We computed the gradient of the image, which produced a vector field from the outside to the inside. We integrated this vector field to produce a series of lines, each for every vertex in the manual cerebellar segmentation. The vector field was integrated only if the grey values decreased, and if a minimum was reached, the vector field was continued linearly. At the end of this procedure, all lines had the same length, computed so as to cover the whole range from the surface of the cerebellum to the granular layer. The grey levels in the original image were sampled along each scan line, producing a grey level profile. The grey level profiles were derived, and a peak detection function was used to determine the point of maximum white/grey contrast (Figure 4c, where the red dot shows the detected boundary), indicating the boundary between the molecular and granular layers. Figure 4b shows the part of the profile lines starting at the cerebellar surface and ending at the detected boundary. For each scan line the corresponding thickness of the molecular layer was defined as the length from the manual cerebellar segmentation until the maximum contrast point (red point). Finally, for each species, a single thickness value was computed as the median of the thickness measured for each profile line, to make our estimation robust to outliers. See the accompanying source code for further implementation details.”

Reviewer #3 (Recommendations for the authors):In general, I am happy with the responses of the authors and the changes they have made. There are still two statistical issues requiring a response:1. The justification for the OU model. Grabowski et al. do not actually demonstrate that the critique by Cooper et al. is misleading – mainly they just show that Cooper et al. were correct! That is not to say that the OU model can never be justified (although some query its basis in biological reality), but that instances of its superiority over BM need to be treated very cautiously. I don't think their simulation solves the problem – the reason the OU is biased is because it does not conform to the data distributions assumptions of AIC (which has strong assumptions about normality or AICc) – this would be the same for other tree transforms like δ (Pagel). This results in the OU being statistically favoured when the 'pull' parameter is so weak it is essentially the same as the Brownian model.

The reviewer’s comment include 2 sub-points:

1) The meaning of Grabowski et al.’s work. The reviewer indicates that Grabowsky et al.’s work mainly shows that Cooper et al. were correct. It is not immediately evident that this is what Grabowsky et al. intended to convey. Referring to Cooper et al.’s work, they write: “we argue that these results have little relevance to the estimation of adaptation with Ornstein–Uhlenbeck models for three reasons.” which they detail. Further in the introduction they write “We disagree, however, with Cooper et al.’s (2016) claim to have detected specific problems with the Ornstein–Uhlenbeck method and we find their recommendations against the application of the method to be unfounded”. The reviewer’s comment points, however, to an important issue, which goes beyond the specific details of Cooper et al.’s and Grabowsky et al.’s papers: that OU models (as well as any other models) need to be appraised critically, and their results not taken at face value. This is especially relevant in the case of our work, which aims at being read by an audience that in many cases will not be familiar with phylogenetic comparative methods. In the revised version of our manuscript we have aimed at making this point more clear, pointing the readers to the relevant literature. We modified our Discussion section as follows:

“Fitting phylogenetic models with a significantly greater number of phenotypes than species is challenging (in particular, there is an ongoing debate regarding the use of OU models, see the work of Cooper et al. 2016, Clavel et al. 2019, and Grabowsky et al. 2023). The use of penalisation techniques and constraints in the model’s structure holds great promise and will enable the exploration of new hypotheses regarding the evolution of complex phenotypes (Clavel et al. 2015, 2019).”

2) Justification of the OU model. We have followed the reviewer’s advice and inspected our selection of the OU model over the BM model at greater detail. We did this using simulations, as suggested in the reviewer’s previous review. Our simulations aimed at replicating the procedure adopted by Cooper et al. – i.e., simulating BM data and evaluating the number of times the OU model is incorrectly selected – while using AICc for model selection which takes model complexity into account (unlike the comparison of log-likelihoods used by Cooper et al). The reviewer comment does not make it clear why our simulations would be incorrect. AIC and AICc criteria are grounded on information theory (they evaluate the amount of information lost by modelling) and do not make distribution assumptions. AIC and AICc require the models to be estimated by maximum likelihood because one can show that asymptotically the deviance plus twice the number of degrees of freedom is proportional to the Kullback-Leibler divergence between models and the observed data, for any probability distribution (Akaike 1974). Moreover, the OU process is a Gaussian diffusion process, and by definition it has a normal distribution. So, if normality were required (which is not the case) that would not violate it. If the “pull” parameter (α in our equations) were close to 0 the OU model would reduce to the BM model and should provide the same fit, however, it would also have more parameters which will result in a higher AIC (AIC = 2k – 2 ln (likelihood)) leading to the selection of the simpler model (BM over OU). Our simulations, which use the same number of variables and species as in our data, show this to be indeed the case. In addition to our simulations, the work of Clavel et al. (2019) provides a thorough exploration of the ability of the penalised likelihood framework implemented by mvMORPH to select the correct model in different scenarios. Finally, as Clavel et al. (2019) show, the instances where one would select OU rather than BM come from upward bias in estimating α, so that we would tend to detect it to be stronger than it is rather than having it shrinking to zero as the reviewer suggests. Again, our aim is not to dismiss the reviewer’s concern, but given the evidence at hand and the results of our analyses, we have no justification for selecting a BM model by principle when both our model selection methodology and our power analysis indicate that we should select the OU model. All our data and code are openly available for criticism, and we make clearer in the revised version of the manuscript that model selection is not without controversy.

“Fitting phylogenetic models with a significantly greater number of phenotypes than species is challenging (in particular, there is an ongoing debate regarding the use of OU models, see the work of Cooper et al. 2016, Clavel et al. 2019, and Grabowsky et al. 2023). The use of penalisation techniques and constraints in the model’s structure holds great promise and will enable the exploration of new hypotheses regarding the evolution of complex phenotypes (Clavel et al. 2015, 2019).”

2. The authors reject the criticism of overfitting by stating that their analyses do not involve fitting models. This is technically correct in strict terms but misleading. Exactly the same issues around small sample sizes per parameter (or variable) arise in partial correlation networks as in fitting a multiple regression model. Partial regression coefficients and partial correlation coefficients are susceptible to the same problem- neither can be estimated accurately without sufficient statistical power. Say you have 20 variables and 20 data points – no one would (hopefully) run a partial correlation on the 20 variables, or even 10 variables, or 5 – maybe just 2 could be justified. Here, the authors have 9 variables and 56 data points – with only 6 data points per variable, they don't have the statistical power to tease apart the relationships as they imply.

We apologise for misinterpreting the reviewer’s previous comment. We have addressed the reviewer’s criticism and performed simulations to estimate our statistical power to detect large, medium and small partial correlations. In general, the assessment of statistical power includes 4 main components: (1) sample size, (2) α value (significance threshold), (3) effect size, and (4) power level. The sample size on its own is not sufficient to estimate statistical power. Neither a sample size of N=20 nor one of N=1,000 are intrinsically small or large: that will depend on the effect size and the α value. N=1,000 may be a large sample size for estimating differences in height between males and females, but would be completely insufficient for a GWAS where effect sizes are infinitesimal. In our case, our effect sizes (the correlations we observe) are extremely large, and our sample size largely sufficient (to clarify, we have a sample of 56 species, with 9 – strongly correlated – variables measured for each species, resulting in 504 data points). We computed our statistical power to detect large (“3 asterisks”, approx. |r|>0.5), medium (“2 asterisks”, approx. |r|>0.34) and small (“1 asterisk”, approx. |r|>0.25) partial correlations. We had over 95% power to detect large partial correlations (α=0.05), over 90% power for medium partial correlations, and over 75% for small partial correlations (see Author response image 1). For this, we generated 1000 simulations of multivariate data following our observed variance-covariance matrix (capturing the correlation structure across our 9 variables) and computed the number of true positives, false positives, true negatives and false negatives. Sensitivity (power) was computed as TP/(TP+FN). We repeated the same procedure for sample sizes going from 20 to 100 species. The reason for this is that the allometric pattern in our data is so strong (PC1 captures >90% of the variance) that large and medium partial correlations could be detected with 80% power even with less than 40 species. Indeed, modifying our observed variance-covariance matrix to weaken the strength of the allometric pattern (PC1) resulted, as expected, in a decreased statistical power level (Author response image 1). The criticism of the reviewer would apply in the limit where all 9 variables would be independent.

**Author response image 1. sa2fig1:** Statistical power for partial correlation analysis. (a) Power to detect the observed effect size was computed using 1000 simulations. At each run a sample size varying from 20 to 100 species was generated with 9 variables using the observed variance-covariance matrix. We estimated our power to detect large, medium and small partial correlations. The simulations show that with N=56 species we had excellent power to detect large and medium partial correlations, and close to 80% power to detect small partial correlations. (b) Power would decrease if the allometry in our data were not as strong as it is, which makes our 9 variables strongly correlated instead of independent. For illustration, we modified the observed variance-covariance matrix to weaken its allometry: we took the square root of all eigenvalues, and scaled the resulting variance-covariance matrix so that its determinant would be the same as in the observed matrix. This results in a less anisotropic, more “spherical” variance-covariance matrix. As expected, statistical power decreases (however, it is still ~90% for large partial correlations). Code for the simulations was uploaded to the accompanying GitHub repository.

We added the statistical power code to our GitHub repository, and the following to the Methods section:

“The statistical significance of each partial correlation being different from 0 was estimated using the edge exclusion test introduced by Whittaker (1990). We evaluated our statistical power to detect partial correlations of different strengths using simulations. Code is included in the accompanying source code along with a Jupyter notebook providing an executable version of our partial correlations example together with further details on our power analysis. A non-executable version of this notebook is provided as Supplementary file 2.”

And to the Results section:

“Our simulations showed that, because of the strength of the allometric pattern in our data, we had excellent statistical power to detect large and medium partial correlations (90-100%) and good statistical power to detect small partial correlations (~78%, see Supplemental Methods for details).”

Added to this, even with large samples, throwing in a bunch of variables without some explicit model that you are testing does indeed result in a correlational salad.

We understand that the reviewer favours a model-based approach to our exploratory approach. Both approaches are, however, equally valid and used, yet with different aims in mind. Here we interpret it as “an unprincipled selection of variables and their bivariate correlations”. First, our selection of variables is justified, and their more thorough presentation was already addressed following the previous comments from reviewers #1 and #2. Briefly, they cover different but complementary aspects of anatomical diversity (body size, brain size, section area and length) and stability (folial width, folial perimeter and thickness of the molecular layer). The first group of variables is widely used in the literature, and their inclusion facilitates comparison (some of these comparisons are the subject of our subsection on “validation”). The second group of variables is more infrequent, however, they are fully justified in theoretical terms, in particular given biomechanical models of folding (as described at length in our manuscript). Second, our methodology for the exploration of the correlation structure of the data is based on well established multivariate methods (principal component analysis, partial correlations analysis) whose aim is exactly to avoid an unprincipled choice of arbitrary bivariate correlations. Our analyses also take phylogenetic structure into account – in a multivariate manner. However, if “correlational salad” refers only to the presence of many correlations, then yes, our multivariate analysis of correlation structure could be seen as many bivariate correlations (although they all result from a single matrix decomposition operation in the case of PCA, and a single matrix inversion in the case of partial correlations), but this is a valid approach for data exploration that is able to provide helpful insight.

Typically, such a partial correlation network would be liable to instability, and the more variables you add the harder it is to make sense of the individual relationships between them.

Our simulations help address this concern, as they show that we have excellent statistical power to detect the partial correlations conveyed by the variance-covariance matrix of our variables. While we acknowledge that in many instances partial correlations may offer unreliable information, in the current scenario this is not applicable due to the exceptionally strong allometric pattern present in our data (strong effect size).

Their explanation of how a negative correlation can arise between variables after controlling for their relationship to other variables is not necessary – this is obvious. What is not so obvious is why in principle there should be a negative (partial) correlation within their network – what is its interpretation? If they started with a model that predicted this or had some compelling explanation, I would be more impressed, but as it is I suspect it is an artefact for the reasons stated above.

We apologise for having misunderstood the previous comment of the reviewer concerning the impossibility to interpret negative partial correlations. Our aim was to provide a simple example showing how they arise and which type of information they may convey. Partial correlation analysis is one of the basic tools for exploratory data analysis, and we believe their general usefulness is beyond doubt.

In the precise case of our results, partial correlation provides interesting perspectives on several aspects. For example, they suggest that the strong correlation between cerebellar and cerebral section area and length is not completely the reflection of a common correlation with body size or brain weight: cerebellar section area would be significantly correlated (Figure 7c, r=0.328) with cerebral section area even if the influence of all other variables were removed. Additionally, the two significant negative correlations (r<-0.34, r<-0.36, p<0.01) are also interesting, as they suggest a decoupling between diverse and stable phenotypes. This information is particularly relevant for the discussion concerning cerebral and cerebellar folding, as they point to a similar folding mechanism for both structures. We do not pretend to provide a definitive answer on the issue, but our results provide a compelling direction for future investigation. The statistical significance of our findings, reported in the previous revision, provided evidence on their robustness, which is now reinforced by our power analysis.

We thank the reviewer for his comments, which have motivated us in particular to perform additional analyses which demonstrate more clearly the validity of our results, while also stating more clearly their potential limitations. The reviewer had concerns about sample sizes, and the pertinence of OU models and partial correlation analyses which we have tried to operationalise and approach through simulations and analyses of statistical power. We hope the exposition of our results is now more clear, as well as the evidence of their robustness.

References

Akaike, H. (1974). A new look at the statistical model identification. In IEEE Transactions on Automatic Control (Vol. 19, Issue 6, pp. 716–723). Institute of Electrical and Electronics Engineers (IEEE). https://doi.org/10.1109/tac.1974.1100705

Clavel, J., Aristide, L., and Morlon, H. (2018). A Penalized Likelihood Framework for High-Dimensional Phylogenetic Comparative Methods and an Application to New-World Monkeys Brain Evolution. In L. Harmon (Ed.), Systematic Biology (Vol. 68, Issue 1, pp. 93–116).

Oxford University Press (OUP). https://doi.org/10.1093/sysbio/syy045

Cooper, N., Thomas, G. H., Venditti, C., Meade, A., and Freckleton, R. P. (2015). A cautionary note on the use of Ornstein Uhlenbeck models in macroevolutionary studies. In Biological Journal of the Linnean Society (Vol. 118, Issue 1, pp. 64–77). Oxford University Press (OUP). https://doi.org/10.1111/bij.12701

Grabowski, M., Pienaar, J., Voje, K. L., Andersson, S., Fuentes-González, J., Kopperud, B. T., Moen, D. S., Tsuboi, M., Uyeda, J., and Hansen, T. F. (2023). A Cautionary Note on “A Cautionary Note on the Use of Ornstein Uhlenbeck Models in Macroevolutionary Studies.” In Systematic Biology. Oxford University Press (OUP). https://doi.org/10.1093/sysbio/syad012